# STimage-1K4M: A histopathology image-gene expression dataset for spatial transcriptomics

**Jiawen Chen**    **Muqing Zhou**    **Wenrong Wu**    **Jinwei Zhang**    **Yun Li**[*]    **Didong Li**[*]

**University of North Carolina at Chapel Hill**
{jiawenn, muqingz, wenrong, jinweizh, yun_li, didongli}@unc.edu

## Abstract

Recent advances in multi-modal algorithms have driven and been driven by the increasing availability of large image-text datasets, leading to significant strides in various fields, including computational pathology. However, in most existing medical image-text datasets, the text typically provides high-level summaries that may not sufficiently describe sub-tile regions within a large pathology image. For example, an image might cover an extensive tissue area containing cancerous and healthy regions, but the accompanying text might only specify that this image is a cancer slide, lacking the nuanced details needed for in-depth analysis. In this study, we introduce STimage-1K4M, a novel dataset designed to bridge this gap by providing genomic features for sub-tile images. STimage-1K4M contains 1,149 images derived from spatial transcriptomics data, which captures gene expression information at the level of individual spatial spots within a pathology image. Specifically, each image in the dataset is broken down into smaller sub-image tiles, with each tile paired with $15,000 - 30,000$ dimensional gene expressions. With $4,293,195$ pairs of sub-tile images and gene expressions, STimage-1K4M offers unprecedented granularity, paving the way for a wide range of advanced research in multi-modal data analysis an innovative applications in computational pathology, and beyond. [1]

## 1 Introduction

Multi-modal data, especially image-text pairs, has gained significant importance and popularity (Srinivasan et al., 2021; Schuhmann et al., 2022), driven by the recent success of multi-modal models such as Contrastive Language-Image Pre-Training (CLIP, Radford et al. (2021)). Researchers have been leveraging these models and data across various fields due to their versatility. Initially, many image models were trained to predict a fixed set of predetermined object categories, limiting their generalizability to identify other visual objects or concepts. Learning directly from text descriptions about images provides a complementary and broader source of supervision, expanding the range of potential applications.

Histopathology plays a crucial role in medical diagnostics, focusing on the microscopic examination of tissue samples to detect diseases and guide treatment decisions (Bera et al., 2019). It helps identify cellular abnormalities, including cancerous cells, inflammation, and tissue degeneration (Reddy, 1996; Bera et al., 2019). The collection of image-text pair data for histopathology requires careful annotation of whole-slide images (Huang et al., 2023; Ikezogwo et al., 2024) to create large-scale

---

[*]Corresponding authors
[1]The data and code are publicly available at https://github.com/JiawenChenn/STimage-1K4M and https://huggingface.co/datasets/jiawennnn/STimage-1K4M

38th Conference on Neural Information Processing Systems (NeurIPS 2024) Track on Datasets and Benchmarks.

datasets suitable for research, training, and diagnostic tool development. Recent efforts to collect and annotate histopathology slides have opened up new opportunities in this domain (Gamper et al., 2019; Graham et al., 2021; Amgad et al., 2019; Huang et al., 2023; Ikezogwo et al., 2024). These annotations vary from simple single labels, such as cell/nuclei types in PanNuke (Gamper et al., 2019) and Lizard datasets (Graham et al., 2021), and cancer regions in NuCLS (Amgad et al., 2019). They also extend to more complex natural language descriptions derived from social media sources such as Twitter or YouTube, as seen in datasets such as OpenPath (Huang et al., 2023) and Quilt-1M (Ikezogwo et al., 2024). Fine-tuning multi-modal models like CLIP with these diverse datasets has shown improved performance in various tasks, including tissue structure classification and image/text retrieval (Huang et al., 2023; Ikezogwo et al., 2024). By combining the capabilities of multi-modal models with detailed histopathology annotations, researchers can achieve greater accuracy and flexibility in medical image analysis. This advancement not only enhances the diagnostic process but also opens the door to new applications in the development of automated pathology tools and, more generally, in personalized medicine (Liu et al., 2019; Nikolov et al., 2021).

While advancements in image annotation have shifted from single labels to natural language descriptions, histopathology slides remain complex and contain a wealth of information that can be challenging to encapsulate in a limited amount of text (Radford et al., 2021; Chen and Zou, 2023). These large tissue slides often feature diverse tissue structures, making it difficult to accurately describe all aspects within a confined-length text. This complexity is further compounded in slides depicting certain diseases, where the focus tends to be on diseased regions, potentially overlooking healthy tissue areas. Randomly cropping images from these slides can lead to misinterpretation and incorrect annotations (Ciga et al., 2021). Histopathology slides are commonly stained with Hematoxylin and Eosin (H&E), revealing details like nuclei and stroma. However, much more biological information exists in these tissue samples, such as gene expression changes and cell-cell communication, which cannot be discerned through staining alone.

Gene expression, the process through which mRNA molecules are generated from the information encoded by the DNA of a gene, is pivotal in studying biological processes. Gene expression data can significantly enhance the annotation of histopathology images. For instance, cancer regions can be identified by the over-expression of specific genes like *ERBB2* in human epidermal growth factor receptor 2 (HER2)-positive breast cancer (Andersson et al., 2021). Moreover, gene expression data can support various downstream analyses, such as deconvolution (Chen et al., 2022, 2023; Luo et al., 2024), which infers the proportion of different cell types in a sample, or clustering (Yuan et al., 2024; Hu et al., 2021; Luecken et al., 2022), which can reveal distinct cell/tissue types/states. The potential applications of gene expression data mark the potential benefit of such paired image and gene expression data.

Gene expression can be measured through several technologies. Bulk RNA-sequencing provides an average expression across large cell populations (Kukurba and Montgomery, 2015). Single-cell RNA sequencing allows for analysis at the individual cell level, enabling more detailed insights into cellular heterogeneity (Kukurba and Montgomery, 2015). However, it still loses the spatial context within the tissue, which is crucial for integrating gene expression data with pathology images to utilize multi-modal methods effectively. To address this need, we highlight spatial transcriptomics (ST) (Ståhl et al., 2016; Moses and Pachter, 2022), a technology that uniquely measures gene expression while preserving spatial information within the tissue (Figure 1a,b). To be more specific, ST can provide gene expression measurement for individual sub-tiles that altogether make up the whole tissue slide. ST has gained significant attention and popularity in recent years due to this unique ability to measure gene expression within spatial context (Ståhl et al., 2016). These ST technologies have revolutionized the way researchers study tissue, allowing for more in-depth analysis of spatial interactions within the tissue and insights into tissue organization and disease mechanisms (Moses and Pachter, 2022; Tian et al., 2023). A key advantage of ST is its ability to provide both high-resolution histopathology images and detailed whole-transcriptome data for each spatial coordinate within a large tissue image (Ståhl et al., 2016). This makes ST a perfect source for paired medical image and text datasets, offering a richer, more accurate annotation that addresses the limitations of over-simplified textual descriptions that typically focus solely on broad categories like cancer or non-cancer regions. By providing high-dimensional annotations for each sub-tile, ST enables a more comprehensive understanding of tissue granularity, facilitating studies of cell-cell communication, tissue architecture, and disease progression (Ståhl et al., 2016; Tian et al., 2023).

Despite these advantages, existing datasets that combine pathology images with gene expression data are often limited in size and scope (Fan et al., 2020; Xu et al., 2022; Fan et al., 2023; Yuan et al., 2023; Zheng et al., 2023; Moses and Pachter, 2022; Zhou et al., 2024; Li et al., 2022), preventing the full utilization of advanced multi-modal models. To bridge this gap, we curated the STimage-1K4M dataset (Figure 1), a comprehensive collection of 1,149 ST slides from three leading ST technologies that provide histopathology images: Spatial Transcriptomics (Ståhl et al., 2016) (note that we use the full name to indicate this particular technology, and ST for general ST technologies), Visium, and VisiumHD. These slides were further subdivided into smaller sub-tiles, resulting in a total of 4,293,195 images, each with corresponding high-dimensional gene expression data. This extensive dataset spans 10 different species and includes 50 distinct tissue types. STimage-1K4M represents a significant advancement in multi-modal datasets for computational pathology and related fields. By providing a large and diverse collection of image sub-tiles paired with detailed gene expression data, the dataset offers researchers a unique resource for exploring the spatial organization of tissues and understanding the complex relationships between cellular structures, gene activity, and disease and health related outcomes. This dataset paves the way for advanced research in multi-modal analysis and innovative applications in computational pathology and personalized medicine.

## 2 Related work

This work builds on existing research in vision-language datasets for histopathology, spatial omics datasets, and representation learning in medical image analysis.

**Vision-Language Pairs in Histopathology.** Multiple histopathology image-text pair datasets have emerged, serving as a foundational resource for studying medical images. The ARCH dataset consists of 8,617 figure-caption pairs with histology or immunohistochemistry (IHC) images, curated from research publications (Gamper and Rajpoot, 2021). The OpenPath dataset offers a broader perspective, featuring 116,504 image-text pairs from Twitter posts across 32 pathology subspecialties, along with 59,869 image-text pairs from replies to popular tweets, and 32,041 additional image-text pairs scraped from the LAION dataset (Huang et al., 2023; Schuhmann et al., 2022). Quilt-1M, a combination of Quilt with datasets from other sources, represents one of the largest vision-language histopathology datasets to date, with over 1 million image-text samples (Ikezogwo et al., 2024). These datasets prove to be valuable resources for training and evaluating models that can understand and correlate textual information with histopathology images.

**Spatial Omics Datasets.** The rise of ST and spatial omics data has spurred the development of various datasets that focus on transcriptomics or other omics data in tissue samples. Notable databases include SpatialDB (Fan et al., 2020), STOmicsDB (Xu et al., 2022), SPASCER (Fan et al., 2023), SODB (Yuan et al., 2023), Aquila (Zheng et al., 2023), Museum of Spatial Transcriptomics (Moses and Pachter, 2022), SORC Zhou et al. (2024), and SOAR (Li et al., 2022). These datasets focus primarily on gene expression data, providing researchers with a wealth of information about the spatial distribution of gene expression in tissue samples. However, there is currently a lack of datasets that provide paired image and gene expression data, which is crucial for bridging the gap between visual information and underlying transcriptomic profiles.

**Representation Learning in Medical Imaging.** Representation learning has made significant strides in medical imaging. Early models focused on predicting single values such as gene expression (He et al., 2020a) or survival outcome (Chen et al., 2021), while more recent approaches employ self-supervised learning (SSL) techniques to learn from unlabeled image data (Ikezogwo et al., 2022). Contrastive SSL models including PLIP (Huang et al., 2023), Quilt-Net (Ikezogwo et al., 2024) and CONCH (Lu et al., 2024), which use image and label annotation, have gained popularity, with models successfully trained on image-text pairs. However, text encoders are limited by token length, making it challenging to incorporate gene expression data. In the ST field, researchers have explored contrastive SSL for image-gene expression data or other modalities like gene expression paired with protein abundance (Zeng et al., 2023; Long et al., 2023; Yao et al., 2024). These models are typically trained on a single slide, constrained by the lack of large datasets that pair histopathology images with gene expression data, and the challenge of integrating gene expression across different datasets.

# 3  Curating STimage-1K4M: Overview

ST technologies can be broadly categorized into two main types: sequencing-based and imaging-based. Sequencing-based ST technology typically involves capturing spatial information using unique barcodes that correspond to specific regions which are usually called "spots" within a tissue sample (see Figure 1a middle panel and Figure 1b for example). This approach enables researchers to capture the entire transcriptome while retaining the spatial context through the barcodes. Imaging-based ST technology, on the other hand, uses fluorescence or other imaging techniques to visualize gene expression directly in the tissue context, and can reach cellular and even sub-cellular resolution. However, imaging-based ST technology has a limitation: the number of genes it can measure is restricted, due to the complexity of multiple rounds of fluorescence of many genes. To be more specific, sequencing-based technologies like Spatial Transcriptomics (Ståhl et al., 2016), Visium and VisiumHD (10x Genomics) can measure ∼15k-30k genes (Figure 1b) while imaging-based technologies like MERFISH (Chen et al., 2015) and STARmap (Wang et al., 2018) can only measure hundreds of genes (median number of genes around 300 in the SOAR database (Li et al., 2022)).

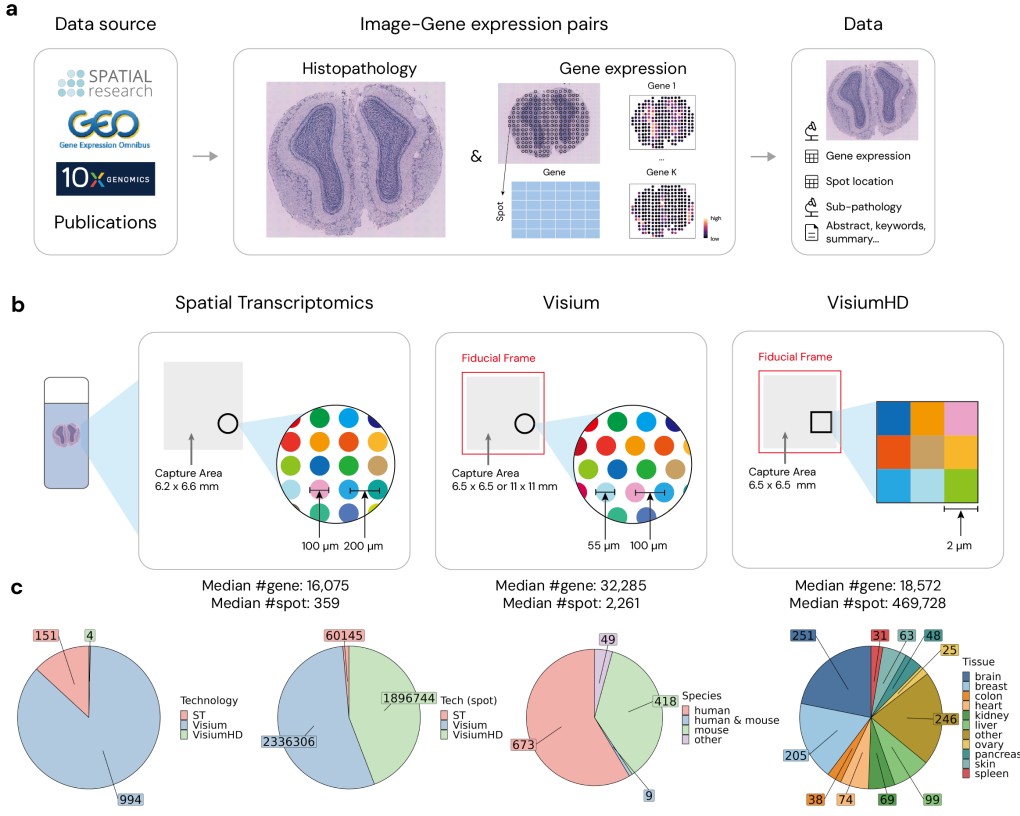

Figure 1: Overview of STimage-1K4M. (a) Curation overview. (b) ST technologies resolution. (c) Breakdown of technologies, species and tissue types in STimage-1K4M.

To obtain comprehensive gene expression information across the entire transcriptome, we focus on sequencing-based ST technologies, specifically those that also offer histopathology images: Spatial Transcriptomics, Visium, and VisiumHD (Figure 1b). Spatial Transcriptomics Ståhl et al. (2016) is one of the first sequencing-based ST methods that allows spatial mapping of gene expression in tissue sections. This technology uses unique barcodes to capture gene expression at specific spots in a grid pattern, with a spot diameter of 100 $\mu m$ and a center-to-center distance of 200 $\mu m$ (Figure 1b left panel). Visium evolves from Spatial Transcriptomics, offering improved resolution and higher throughput with a spot diameter of 55 $\mu m$ and a refined center-to-center distance of 100

$\mu m$ (Figure 1b middle panel). VisiumHD further extends the capabilities of Visium by offering even finer resolution and data density. This advanced technology introduces a continuous measurement system, utilizing a $2\mu m \times 2\mu m$ grid of bins for gene expression analysis (Figure 1b right panel). In our work, we employed an $8\mu m \times 8\mu m$ bin structure, as instructed by 10X Genomics, achieved by aggregating the gene expression data from smaller $2\mu m \times 2\mu m$ bins to match the desired resolution.

Public available sources for ST data include Gene Expression Omnibus (GEO), 10X Genomics datasets, Spatial Research datasets, and various publications. We queried the GEO website using keywords "spatial transcriptomics", specifically targeting supplementary files in JPG, PNG, or TIFF formats. This search resulted in 856 datasets from 121 unique GEO studies. Additionally, we gathered 58 Visium and 4 VisiumHD datasets from 10X Genomics, complementing these with 233 slides manually collected from 10 additional studies (see Appendix A for a full list of references).

A significant challenge in this process was the inconsistent sharing standards for ST data, particularly for the image components. Many datasets lack corresponding images, making it difficult to analyze the gene expression data in its proper spatial context. For Visium data, the standard format typically includes at least one image, which can be of full-resolution, high-resolution, and low-resolution. In this work, we used the highest resolution images available for each dataset. Spatial Transcriptomics data posed additional hurdles. This kind of data requires CytAssist images to map the coordinates to the image, but these images are rarely publicly available, making it challenging to link gene expression data to histopathology images. Only datasets with mapped and unmapped coordinates could be included in the study for the calculation of spot diameter following ST pipeline in the SpatialTranscriptomicsResearch GitHub repository. Given the various sharing formats and the common absence of key data, it's particularly challenging for researchers unfamiliar with ST to align gene expression data with histopathology images. To address this, we manually processed and verified every dataset to ensure accurate coordinate mapping, allowing precise linking of gene expression data to histopathology images. Furthermore, we calculated and included the corresponding spot radius to indicate the area of measurement. These manual efforts underscore our commitment to providing a reliable and comprehensive dataset, facilitating easier integration of ST data with histopathology images for researchers across various disciplines.

In summary, we systematically collected a diverse collection of 1,149 ST slides, encompassing 4,293,195 spots with paired gene expression information. For each dataset, we provide histopathology images, spot center coordinates and radius, as well as the associated gene expression data. Our STimage-1K4M dataset comprises of data from Spatial Transcriptomics, Visium, and VisiumHD platforms. At the slide level, STimage-1K4M has 13.1% from Spatial Transcriptomics, 86.5% from Visium, and 0.3% from VisiumHD. At the spot level, due to the resolution difference, the composition shifts to 1.4% from Spatial Transcriptomics, 54.4% from Visium, and 44.2% from VisiumHD. STimage-1K4M predominantly includes data from human and mouse, encompasses 50 tissues with the largest proportion of images from brain, accounting for 21.8% (251 slides), followed by the breast tissue at 17.8% (205 slides). Given a major focus on cancer in the field of ST, it's noteworthy that 39.7% of the slides (456 slides) are from studies related to cancer.

In addition to the paired image and gene expression data, we also included pathologist annotations for the slides (Figure 6). Spatial domain detection or clustering is a popular topic in ST data analysis. However, due to the lack of organized datasets, evaluations in most ST clustering methods utilizing image data rely on limited samples (Andersson et al., 2021; Maynard et al., 2021). We manually reviewed relevant publications and extracted annotations from 9 studies including 71 slides to enrich our STimage-1K4M dataset. These pathologist annotations are anticipated to substantially reduce efforts required for collecting labeled data with "ground truth" in the ST field and to provide researchers with a more comprehensive resource for evaluating clustering methods and dimension reduction techniques.

As a comprehensive and meticulously curated dataset, STimage-1K4M aims to facilitate research in ST, computational pathology, and related fields. This dataset can significantly streamline the data collection process, allowing researchers to focus on developing innovative methods and gaining deeper insights into tissue structure and gene expression patterns.

# 4 Popular tasks using ST images

Within the field of traditional computational biology with no gene expression involved, commonly performed tasks such as tissue type classification and image-text retrieval have well-established solutions. However, ST introduces new complexity and opportunity with additional gene expression information. ST data allows researchers to engage in a variety of specialized tasks that are particularly suited to the strengths of this new type of technology.

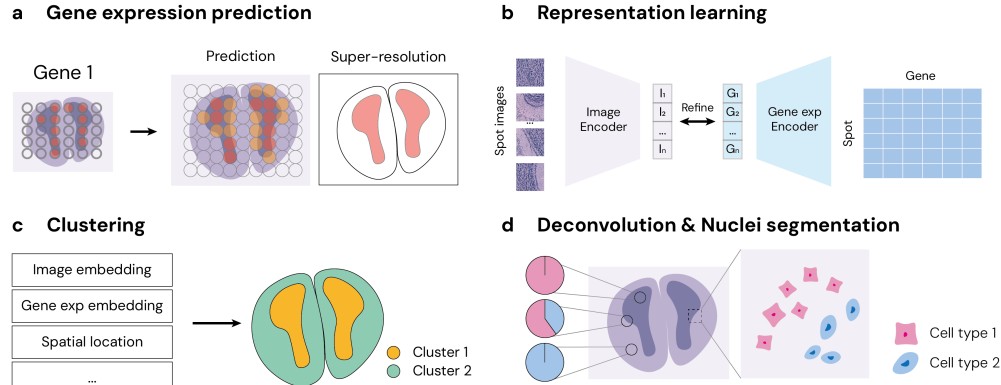

Figure 2: Popular tasks in ST data analysis.

**Gene Expression Prediction and Resolution Enhancement.** One key usage of images in ST is predicting gene expression (Figure 2a) from histopathology images (Xie et al., 2024; He et al., 2020a). This approach allows researchers to infer gene expression levels from visual data, potentially reducing the need for expensive and time-consuming library preparation and sequencing. Additionally, increasing the resolution of gene expression data through high-quality imaging techniques offers a more detailed understanding of spatial patterns within tissue samples, leading to improved accuracy in analyzing gene expression spatial distributions (Hu et al., 2023; Zhang et al., 2024).

**Representation Learning and Clustering.** Similar as in image-based computational biology, learning image embeddings is also a popular task in ST (Figure 2b). This process involves transforming high-resolution tissue images into compact, informative representations that capture the essential features of the underlying biological processes. A key application of these embeddings is spatial clustering (Figure 2c), where similar tissue regions are grouped based on shared characteristics captured in the embeddings (Hu et al., 2021). Clustering allows researchers to explore tissue heterogeneity and identify distinct spatial clusters that may correspond to different cellular functions or disease states.

**Deconvolution and Cell Segmentation.** Deconvolution and cell segmentation are valuable computational methods that enhance our understanding of tissue composition at cellular level (Figure 2d). Deconvolution specifically focuses on deciphering mixed signals within spot-level gene expression data to accurately estimate the proportions of contributing cell types present in a tissue sample. Histology images are particularly valuable in this context because common staining methods inherently highlight nuclei information, providing a clear visual representation for cellular structures (Biancalani et al., 2021; Chen et al., 2023). This visual clarity allows deconvolution via computational methods, as spots that appear similar in the images are likely to have similar cell type compositions. Additionally, the integration of image analysis with deconvolution facilitates the application of trained models to new images or to areas within images where spots were not initially measured, potentially increasing analysis resolution. Furthermore, by employing cell segmentation techniques alongside these images, researchers can precisely identify and categorize individual nucleus, which allows accurate assignment of specific cell types to these identified nucleus, thereby enriching the gene expression data with detailed cellular annotations (Biancalani et al., 2021; Zhang et al., 2024).

# 5 Experiment training with STimage-1K4M

The STimage-1K4M dataset presents unprecedented opportunities for benchmarking, discovery, and training new models. In this section, we provide several examples illustrating its potential uses.

## 5.1 Gene expression prediction

We initially utilize the STimage-1K4M dataset to benchmark the task of gene expression prediction (Figure 2a). With each spot image paired with high-dimensional gene expression data, STimage-1K4M serves as an ideal resource for both learning gene expression and benchmarking large vision models. We assess the performance of image encoders from the pre-trained CLIP, PLIP, and UNI models using the HER2+ human breast cancer dataset. *ERBB2*, a marker gene for HER2+ breast cancer, was selected for this analysis (Figure 3a,b). We trained a simple regression model to predict ERBB2 gene expression using the image embeddings from each model. The PLIP model demonstrated superior performance in this context, exhibiting higher correlation and lower mean squared error (MSE) (Figure 3).

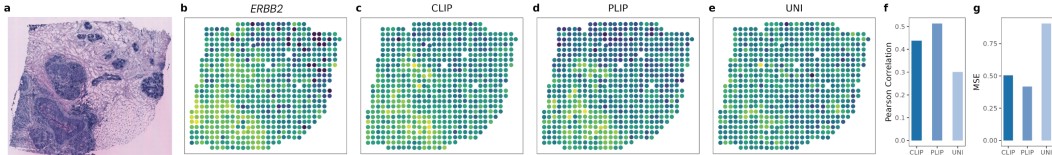

Figure 3: Gene expression prediction evaluation. (a) H2 histology images. (b) *ERBB2* gene expression. (c) Predicted *ERBB2* expression using zero-shot CLIP image embedding. (d) Predicted *ERBB2* expression using zero-shot PLIP image embedding.(e) Predicted *ERBB2* expression using zero-shot UNI image embedding. (f) Pearson Correlation of three models. (g) MSE of three models.

## 5.2 Representation learning and clustering

To demonstrate the effectiveness of our STimage-1K4M dataset, we employed contrastive learning to fine-tune the image encoders of pre-trained CLIP and PLIP models using STimage-1K4M, to enhance the models' performance in integrating pathology images with corresponding gene expressions. To effectively incorporate gene expressions, we replaced the text encoder in these models with fully connected neural networks, as shown in Figure 2b and Appendix B. The objective of our contrastive learning remains consistent with the original CLIP framework, aiming to increase the cosine similarity between embeddings of aligned pairs while minimizing similarity for unaligned pairs.

Given the challenges of different genes measured across datasets and prevailing batch effects, we limited our analyses to samples from Maynard et al. (2021), which includes 12 human dorsolateral prefrontal cortex (DLPFC) slides encompassing 47,681 spots. To manage the high dimensionality of gene expression data, we explored two strategies: highly variable genes (HVG) selected separately from each slide, and HVGs selected from overlapping genes across slides (overlap-HVG). Once fine-tuned, we conducted experiments for image classification using linear probing and analyzed the image embeddings through t-Distributed Stochastic Neighbor Embedding (t-SNE) (Van der Maaten and Hinton, 2008). See Appendix for experiment details.

**Evaluation using linear probing.** We evaluated the performance of the fine-tuned models via linear probing. As shown in Figure 4a, the fine-tuned CLIP and PLIP with HVG achieved higher mean F1 scores that zero-shot CLIP and PLIP models, indicating that fine-tuning on our STimage-1K4M improves the performance. While we did not fine-tune the larger UNI model due to computational constraints, the results suggest that both fine-tuning with our dataset and using a more effective pre-trained model contribute to better performance. We conjecture that fine-tuning UNI on our STimage-1K4M could further enhance its performance, combining the benefits of both advanced model architecture and tailored training data.

**Image representation learning.** To evaluate the enhancement in image representations achieved by the fine-tuned models, we utilized pathologist-annotated brain layers (Figure 4c) as benchmarks to calculate several cluster quality metrics (Figure 4b), including the Silhouette score (Rousseeuw, 1987), the Calinski-Harabasz index (Caliński and Harabasz, 1974), and the Davies-Bouldin index (Davies and Bouldin, 1979). Additionally, we applied t-SNE (Van der Maaten and Hinton, 2008) for visaulization to further analyze the clustering patterns(Figure 4d). Our findings indicate that, compared to zero-shot embeddings, the fine-tuned embeddings more effectively distinguish between various tissue subtypes, notably between white matter (WM) and other layers (L1–L6) in the brain

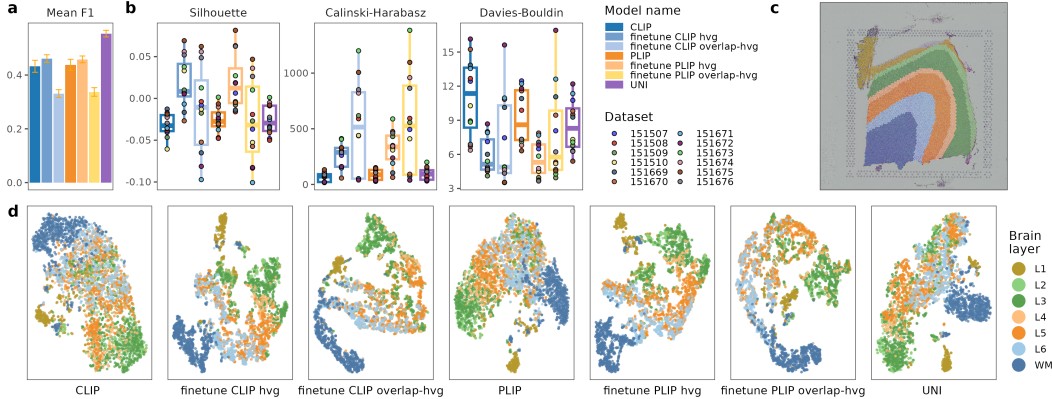

Figure 4: Evaluation results. (a) Linear probing results, denoted by average macro F1 (error bars indicate standard deviations). (b) Silhouette, Calinski-Harabasz and Davies-Bouldin scores for image embeddings. (c) Histopathology image of brain sample 151675 colored by pathologist annotation. (d) t-SNE embeddings of sample 151675, colored by the same layer annotations as in (c).

(Figure 4b,d). In particular, the image embeddings from the fine-tuned models outperform all zero-shot image embeddings. This enhancement suggests that incorporating gene expression data into the training process helps the model capture more nuanced differences within the tissue slides, which highlights the potential of integrating genetic and image information to learn more precise and informative interpretations of tissue structure and function.

### 5.3 Deconvolution and cell segmentation

In ST, a conventional approach for the nuclei segmentation and deconvolution task involves spot-level data. This includes gene expression, spatial coordinates, and image data, often supplemented by an external scRNA-seq reference to estimate cell type proportions for each spot. These inferred cell type proportions could serve as a benchmark, offering an alternative to traditional nuclei and cell type annotations which typically require extensive manual effort. The STimage-1K4M dataset allows for the benchmarking of cell segmentation methods against these inferred cell type proportions.

We evaluated the performance of the CellViT-256 and CellViT-SAM (Hörst et al., 2024) models in segmenting Neoplastic cells using HER2+ breast cancer data, comparing these results with the cancer epithelial cell proportions inferred by a computational deconvolution method RCTD (Cable et al., 2022). Visual inspection supports the models' comparable effectiveness in cell segmentation. Notably, the CellViT-SAM model tends to classify more cells as Neoplastic. We employed Pearson correlation to measure the agreement between the segmented and inferred cell type proportions, with the CellViT-256 model demonstrating a higher correlation (0.37 compared to 0.34 for the CellViT-SAM) (Figure 5).

## 6   Discussion

In this work, we introduced STimage-1K4M, a groundbreaking open-source dataset that pairs histopathology images with gene expression data. Our empirical results demonstrate the effectiveness of pre-training using STimage-1K4M, which has shown to outperform larger state-of-the-art models such as CLIP and PLIP. This success highlights the significant potential of integrating image and gene expression data to enhance model performance and provide new opportunities for advancing research in ST and computational pathology. For further reading on the innovative computational techniques being explored in this field, readers may refer to Szałata et al. (2024); Coleman et al. (2024). Despite these promising results, this emerging field also presents significant challenges that require innovative approaches to overcome. Next, we discuss the potentials and challenges associated with this integration.

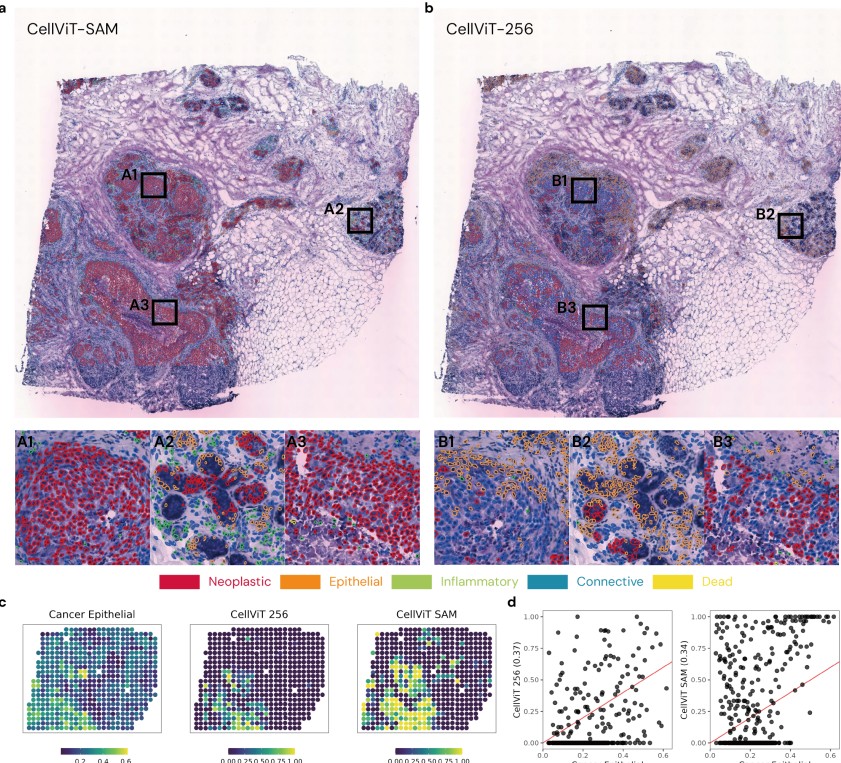

Figure 5: Cell segmentation evaluation. (a) CellViT-SAM model segmented cells. (b) CellViT-256 model segmented cells. (c) left: RCTD inferred cancer epithelial cell proportion in spot leve., middle: aggregated CellViT-256 segmented neoplastic cell proportion. Right: aggregated CellViT-SAM segmented neoplastic cell proportion. (d) scatter plot of RCTD inferred cell type proportion and aggregated segmented cell proportion. Ylab: model name (Pearson correlation).

**High-dimensional image.** A typical histopathology image consists of three primary color channels — red, green, and blue (RGB). These channels represent the standard visualization used in most imaging technologies to capture the visual structure and patterns in tissue samples. When histopathology images are paired with gene expression data, the data dimension increases enormously, presenting both challenges and opportunities for analysis. If gene expression data is treated as a separate set of "channels", where each gene's expression level is represented as a gray-scale image channel, the entire histopathology image transforms into a high-dimensional data structure. Instead of having just three RGB channels, the transformed image would now have around ∼20,000 channels, each representing the expression of a different gene. This high-dimensionality adds molecular information to the visual data, offering insights far beyond what can be revealed by staining methods.

Given this expanded data structure, a crucial question arises: How can this high-dimensional data be effectively analyzed and utilized? One of the central challenges is to strike an optimal balance between sample size and resolution. When focusing on spot-level images, there's a risk of losing spatial connections between the spots. On the other hand, slide-level information provides a broader context but at the cost of reduced sample size, which could limit the scope of analysis. This challenge leads to further questions: How can slide-level information improve the image embeddings for spot-level images? Can the data be augmented by pairing it with datasets containing image-text or purely image-based information?

For spot-level data, several approaches have been attempted, including contrastive SSL (Long et al., 2023; Yao et al., 2024), but these approaches typically concentrate on spot-level images from a single slide, limiting their generalizability. Determining the optimal approach for analyzing multi-dimensional datasets remains an open question. Should researchers employ contrastive SSL, where models learn from paired image-gene expression data, or treat the dataset as a multi-channel image, where traditional image-processing techniques can be applied? These questions are central

to the ongoing evolution of computational pathology, as they determine the effectiveness of latent embedding extraction and ultimately influence models' performance in real-world applications.

**Position encoding.** In traditional vision transformer models (Dosovitskiy et al., 2020), positional encoding is used to provide context about the relative or absolute positions of input image patches within a sequence. This is crucial because transformers, unlike convolutional neural networks, do not inherently retain information about the order or spatial arrangement of their inputs. Positional encoding typically involves adding a set of coordinates or numerical values to the model's inputs, enabling the model to understand spatial relationships and preserve structure during analysis. In the context of integrating histopathology images with gene expression data, gene expression data could potentially serve as a unique form of positional encoding. By linking specific regions within an image to their corresponding transcriptomic information, researchers can create spatially-aware models that can learn from both visual and transcriptomic cues.

**Gene expression annotation.** Gene expression data has become an indispensable resource in the annotation of complex biological datasets. It has been widely used for various downstream analysis including clustering, which classify cells/spots into distinct groups based on their gene expression profiles, and deconvolution. Recent advancements include integrating large language models (LLMs) to extract meaningful gene expression embeddings (Chen et al., 2023; Schaar et al., 2024), which utilize gene names and text descriptions to enhance data interpretation. However, several significant challenges remain. Variation in genome structures across different species complicate cross-species analysis, and batch effects introduce systematic biases in gene expression measurements. Additionally, using gene names with rankings (Chen and Zou, 2023; Schaar et al., 2024) lacks the precision of quantitative values, and high-dimensionality necessitates effective dimension reduction techniques. Current methods, such as using PCs or HVGs, often faill short in multi-slides analysis across different tissue and species.

Our STimage-1K4M dataset has the potential to address these challenges by providing a large, diverse collection of paired histopathology images and gene expression data across multiple species and tissue types. This dataset may facilitate the development of robust annotation methods that manage high-dimensional data and mitigate batch effects. Additionally, integrating STimage-1K4M with other large-scale datasets, such as those containing only images or only gene expression data (Schaar et al., 2024), could further enhance multi-modal learning and improve our understanding of complex biological processes and disease mechanisms.

**Limitations.** In this work, although we have shown that the integration of gene expression data has enhanced the performance of the pre-trained CLIP and PLIP image encoders, the fine-tuned models are still inferior to the UNI model, suggesting that employing a more powerful foundational model could potentially yield further improvements. However, due to limited computational resources, we were unable to fine-tune the UNI model. Additionally, we only utilized a maximum of 128 dimensions, compressed into a 32-dimensional latent layer, to incorporate gene expression data. This simplistic implementation may not fully capture the complexity and richness of gene expression information. Efforts to fine-tune the models using data from other tissue types, resulted in sub-optimal performance. This suggests that batch effects across datasets introduce noise and variability, significantly impacting model performance. For instance, the tSNE embedding of samples from 12 human brain slides in the Maynard et al. study illustrates significant batch effects. In this visualization, points are clustered by data source rather than by biological layers, clearly highlighting the presence of batch effects (Figure 10). Furthermore, our study has not utilized the slide-level abstract and experimental design text information available in our dataset, which represents another area for potential exploration and application.

**Data collection and societal biases.** STimage-1K4M may exhibit inherent biases due to existing ST publications focusing disproportionately focus on brain tissues and breast cancer samples. Such ascertainment biases introduced by published ST work can influence the outcomes of analyses conducted using our STimage-1K4M dataset, potentially leading to models that are better attuned to recognizing patterns specific to these tissues.

## Acknowledgments and Disclosure of Funding

This work is supported by `R01 AG079291, R01 MH125236, U01 HG011720, P50 HD103573, R56 LM013784, R01 HL149683, and UM1 TR004406`. We thank Dr. Hongtu Zhu for the computational resource support.

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

# Appendix

## A    Studies included

Here we include the full list of reference of studies included in STimage-1K4M: (Ji et al., 2020; Chen et al., 2023; Andersson et al., 2021; He et al., 2020b; Ståhl et al., 2016; Asp et al., 2019; Hasel et al., 2021; Ratz et al., 2022; Lebrigand et al., 2023; Joglekar et al., 2021; Mikheenko et al., 2022; Stein et al., 2022; Prjibelski et al., 2023; Bashkirova et al., 2023; Sanchez-Ferras et al., 2021; Parigi et al., 2022; Tower et al., 2021; Meylan et al., 2022; Ni et al., 2022; Kadur Lakshminarasimha Murthy et al., 2022; Foster et al., 2021; Sudmeier et al., 2022; Hudson and Sudmeier, 2022; Tower et al., 2022; Rustagi et al., 2022; Dixon et al., 2022; Lake et al., 2023; Mohammadi et al., 2023; Yamasaki et al., 2022; Chen et al., 2022; Russ et al., 2022; Misra et al., 2021; Guilliams et al., 2022; Habenicht et al., 2022; Dhainaut et al., 2022; Ren et al., 2023; Kenney et al., 2023; Mitamura et al., 2023; Olaniru et al., 2023; Heezen et al., 2023; Topchyan et al., 2022; Filipescu et al., 2023; Vanrobaeys et al., 2023; Kasmani et al., 2023; Castranio et al., 2023; Barkley et al., 2022; Eum et al., 2024; Canela et al., 2023; Garbarino et al., 2023; Caetano et al., 2023; Tung et al., 2023; Heimli et al., 2022; Arora et al., 2023; Chen et al., 2023; Mauduit et al., 2022; Bassiouni et al., 2023; Lyubetskaya et al., 2022; Lee et al., 2023; Foster et al., 2022; Subramanian et al., 2024; Gu et al., 2022; Coutant et al., 2023; Sanders et al., 2022; Akiyama et al., 2023; Yoshitake et al., 2024; Ballester Roig et al., 2023; Villemin et al., 2023; Caronni et al., 2023; Sukhanov et al., 2023; Zhi et al., 2024; Kerzel et al., 2023; Mirzazadeh et al., 2023; Vanrobaeys et al., 2023; Liu et al., 2023; Chaker et al., 2023; Deshpande et al., 2023a,b; Stec et al., 2023; Chitturi et al., 2023; Pavel et al., 2023; Wu et al., 2024; Moeyersoms et al., 2023; Rauber et al., 2024; Sans et al., 2023; Gharaie et al., 2023; Mei et al., 2023; Nakata et al., 2023; Pham et al., 2023; Andrews et al., 2024; Di Marco et al., 2023; Kawai et al., 2023; Zhong et al., 2023; Lequain et al., 2023; Yu et al., 2024; Ng et al., 2024; Cherief et al., 2023; Rahimikollu et al., 2024; Lowe et al., 2024; Cortese et al., 2023; Huuki-Myers et al., 2023; Maynard et al., 2021; Wu et al., 2021; Kuppe et al., 2022; Erickson et al., 2022). We note here that there are some studies with no valid pmid. The full list of study names will be available at GitHub repository.

We also include pathologist annotation for 71 slides from 9 datasets involving human brain, human breast, human prostate, human kidney, and mouse brain (Figure 6).

## B    CLIP and PLIP finetuning details

To evaluate the potential of STimage-1K4M, we fine-tuned the image encoder part of CLIP (Radford et al., 2021) and PLIP (Huang et al., 2023) model. All model implementations are built upon the training code (issue #83) posted in the CLIP GitHub repository. The hyperparameters are chose to be the same of CLIP training as detailed in Table 1. All the parameters are transformed into fp32. For CLIP, we loaded the pretrained parameters (ViT-B/32) from *openai/clip-vit-base-patch32* from hugging face. For PLIP, we loaded the pretrained parameters (ViT-L/14) from *vinid/plip* from hugging face. In our model architecture, the image encoder feeds into a fully connected layer that reduces its output to a 32-dimensional latent space. Similarly, the gene expression encoder also consists of a single fully connected layer that compresses its high-dimensional input down to a 32-dimensional embedding (Figure 7). These 32-dimensional representations from both the image and gene expression encoders are then utilized as the basis for our contrastive loss function. We fine-tuned the models for 15 epochs. All experiments are performed on single NVIDIA A100 GPU.

To compare the choice of gene sets, we employed two methods.

1. **HVG** For each dataset, we selected the top 128 HVGs. Then the HVGs of all dataset are sorted by highly variable rank and combined regardless of gene names as training data for fine-tuning. The HVG are selected using python scanpy package scanpy.pp.highly_variable_genes default settings (Lause et al., 2021; Wolf et al., 2018).

2. **Overlap HVG** For each training data, we first combined the gene expression with respected to gene names and only keep the overlapping genes. Then we select the top 100 HVGs for the combined gene expression as training data. The HVG are selected using python scanpy package scanpy.experimental.pp.highly_variable_genes default settings (Satija et al., 2015; Wolf et al., 2018).

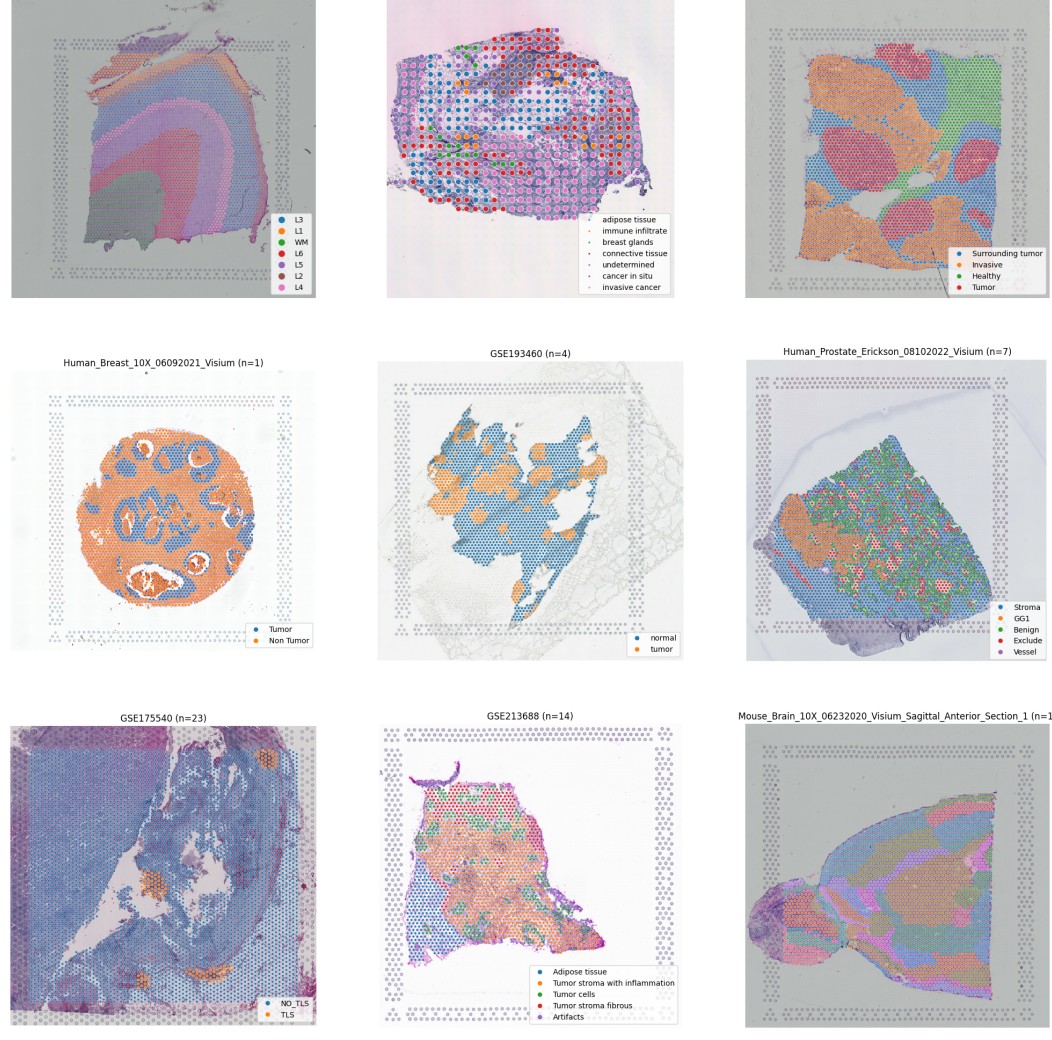

Figure 6: Datasets with pathologist annotation. The points are colored by annotation in each dataset. The legend for mouse brain data (bottom right) are omitted for visualization.

As discussed in Section 6, the gene names of different species are not overlapping. Thus, in order to use combined gene expression from mulitple datasets, we subset STimage-1K4M by species. In this study, we fine-tuned models using all data from (Maynard et al., 2021) (human brain) with different gene sets.

Table 1: CLIP and PLIP fine-tuning hyperparameters.

| Hyperparameter | Value |
|---|---|
| Batch size | 1024 |
| Training epochs | 15 |
| Weight decay | 0.2 |
| Adam $\beta_1$ | 0.9 |
| Adam $\beta_2$ | 0.98 |
| Adam $\epsilon$ | 1e-06 |

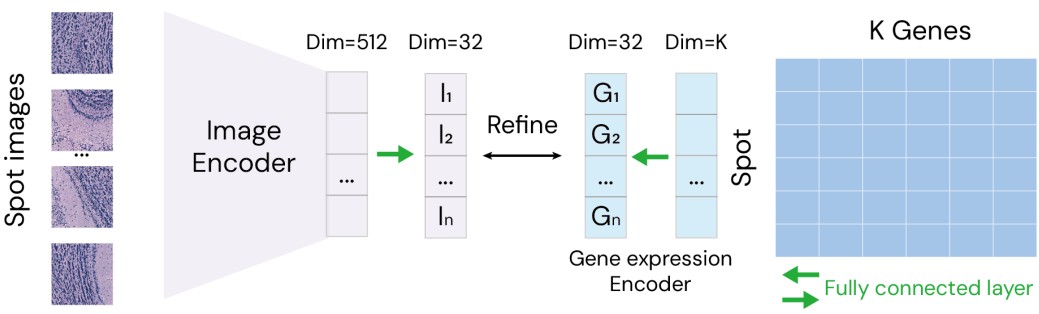

Figure 7: Fine-tune model structure. We replaced the text encoder in PLIP and CLIP with a single fully connected layer (gene expression encoder). We also add a fully connected layer after the image encoder to make the dimension same as the gene expression encoder, which in our case is 32.

## C   Linear probing details

For linear probing, we follow the procedure in (Huang et al., 2023). We employ the stochastic gradient descent classifier (SGDClassifier) module for logistic regression classifier from the sklearn (Pedregosa et al., 2011) Python package. For fine-tuned models, we used 32-dimension image embeddings. For zero-shot models, we used 512-dimension emebddings. The performance of the trained linear classifier using L2 regularization with different regularization multipliers ($\alpha = 1, 0.1, 0.01, 0.001, 0.0001$) are evaluated on the validation splits for all models (training:validataion:test $= 8 : 1 : 1$). The best-performing linear classifier was selected based on the average macro F1 performance from the results trained on training splits sampled by five different random seeds (seed $= 1, 2, 3, 4, 5$). We note here that we fine-tune the models using data from other tissue types, resulted in sub-optimal performance (Figure 8). We also evaluated the performance of the fine-tuned model using the second-last layer image embeddings (512-dimension), the results remain similar as the performance of using 32-dimension image embedding (Figure 9).

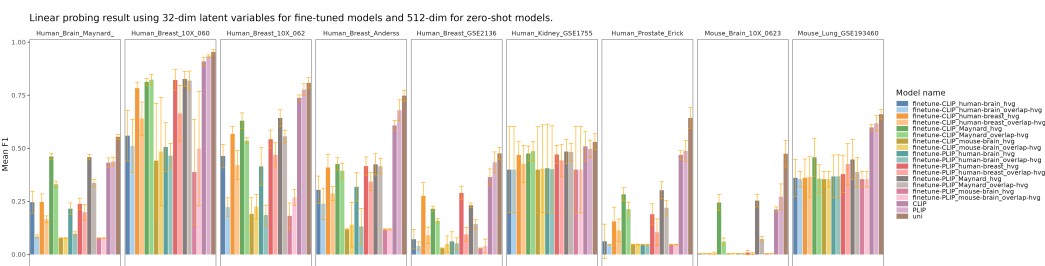

Figure 8: Classification results evaluated on multiple dataset for model fine-tuned with various training data. We fine-tuned the CLIP and PLIP models with human brain, human breast, Maynard et al., mouse brain. For fine-tuned models, we evaluate the 32-dimensional image embedding in a classification task. For zero-shot models, we evaluate the 512-dimensional image embedding. The classification performance of these models are tested on 9 datasets with annotation. The bars are colored by model type. The model name of the fine-tuned models is in "fine-tune model-type training-datatype gene-set" format.

## D   t-SNE details

We applied t-SNE (Van der Maaten and Hinton, 2008) to each slide individually using sklearn.manifold.TSNE function from the sklearn (Pedregosa et al., 2011) Python package. For the fine-tuned models, we utilized 32-dimensional embeddings, whereas for the zero-shot models, we employed 512-dimensional embeddings. We also used this setting in our calculations of the

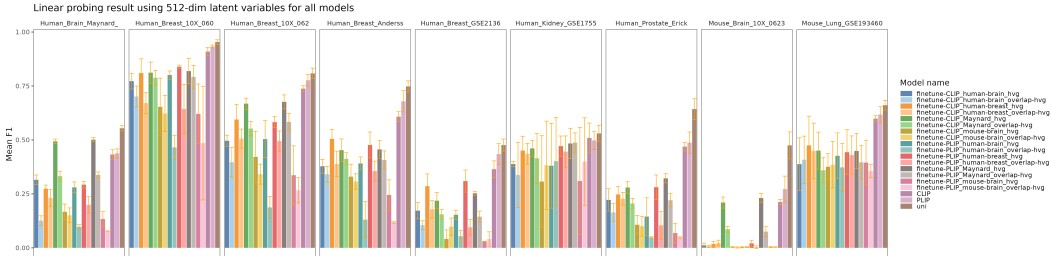

Figure 9: Classification results evaluated on multiple dataset for model fine-tuned with various training data. We fine-tuned the CLIP and PLIP models with human brain, human breast, Maynard et al., mouse brain. For both fine-tuned models and zero-shot models, we evaluate the 512-dimensional image embedding. The classification performance of these models are tested on 9 datasets with annotation. The bars are colored by model type. The model name of the fine-tuned models is in "fine-tune model-type training-datatype gene-set" format.

Silhouette, Calinski-Harabasz, and Davies-Bouldin scores using the sklearn (Pedregosa et al., 2011) Python package. We also included a t-SNE embedding of all samples from the Maynard et al. study, which comprises data from 12 human brain slides and demonstrates notable batch effects.

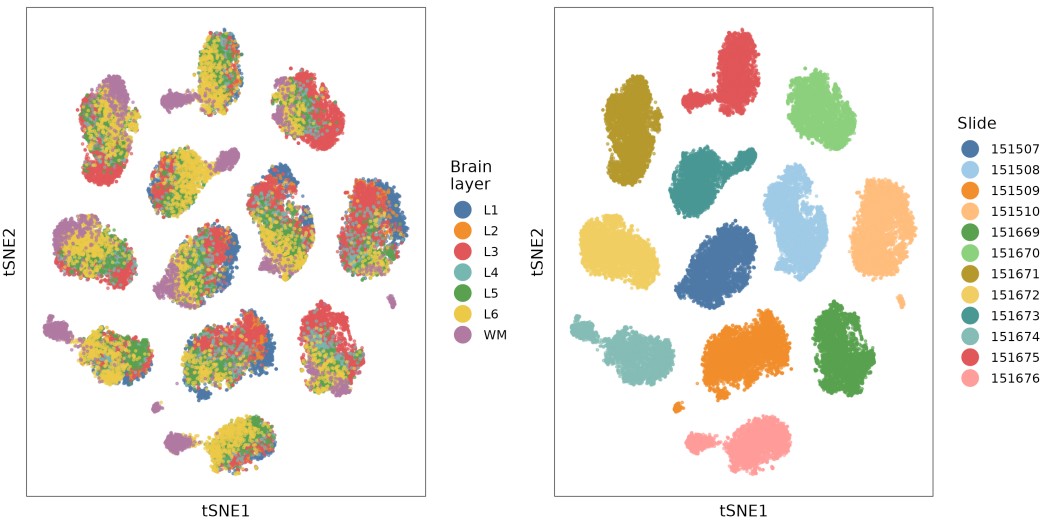

Figure 10: Batch effect in Maynard et al. human brain gene expression with hvg gene set. Left: gene expression tSNE embedding colored by brain layers. Right: gene expression tSNE embedding colored by slide name.

