# Supplementary document for "STimage-1K4M: A histopathology image-gene expression dataset for spatial transcriptomics"

**Jiawen Chen**    **Muqing Zhou**    **Wenrong Wu**    **Jinwei Zhang**    **Yun Li**[*]    **Didong Li**[*]

**University of North Carolina at Chapel Hill**
{jiawenn, muqingz, wenrong, jinweizh, yun_li, didongli}@unc.edu

## S1    DataSheet for STimage-1K4M

We present a DataSheet (Gebru et al., 2021) for STimage-1K4M in this section.

1. Motivation

   - **For what purpose was the dataset created?** STimage-1K4M was created for training histopathology multi-modal, self-supervised models, for understanding pathology images and gene expression data.
   - **Who created the dataset (e.g., which team, research group) and on behalf of which entity (e.g., company, institution, organization)?** This dataset was curated by Dr. Yun Li and Dr. Didong Li's group on behalf of University of North Carolina at Chapel Hill.
   - **Who funded the creation of the dataset?** This work was supported by R01 AG079291, R01 MH125236, U01 HG011720, P50 HD103573, R56 LM013784, R01 HL149683, and UM1 TR004406.

2. Composition

   - **What do the instances that comprise the dataset represent (e.g., documents, photos, people, countries)?** This dataset includes histopathology images, spatial coordinates for spots and the paired gene expressions.
   - **How many instances are there in total (of each type, if appropriate)?** STimage-1K4M includes 1,149 whole-slide images and 4,293,195 spots (sub-tiles) and the expression of 15,000-30,000 genes associated with each spot.
   - **Does the dataset contain all possible instances or is it a sample (not necessarily random) of instances from a larger set?** The STimage-1K4M dataset is not a subset of a larger collection but rather an extensive compilation of all available instances we could identify and collect. We made a comprehensive effort to collect as much data as possible, specifically targeting datasets from the Gene Expression Omnibus (GEO) that contain both pathology images and gene expression data. While it may not cover every possible instance due to the inherent limitations of data availability and access, it represents the most exhaustive collection of such data currently available. We are committed to updating the dataset as more data or technologies become available.

---

[*]Corresponding authors

- **What data does each instance consist of?** We consider each spot as an instance, which has high dimensional gene expression data, image data and spot coordinate data.
- **Is there a label or target associated with each instance?** Yes, the gene expression data could be treated as label for each image.
- **Is any information missing from individual instances?** We provide extra information like abstract, paper title. Such information is missed in datasets without a valid publication id.
- **Are relationships between individual instances made explicit (e.g., users' movie ratings, social network links)?** Yes, all instances of the same slide/dataset and their spatial relationship could be analyzed using the spatial coordinate file.
- **Are there recommended data splits (e.g., training, development/validation, testing)?** There are no recommended data splits, but potentially the data could be split by tissue type.
- **Are there any errors, sources of noise, or redundancies in the dataset?** While the STimage-1K4M dataset is carefully curated, the source ST data may contain inherent noise and errors due to the limitations of the technology used. We are not aware of any redundancies in the dataset.
- **Is the dataset self-contained, or does it link to or otherwise rely on external resources (e.g., websites, tweets, other datasets)?** The dataset is self-contained.
- **Does the dataset contain data that might be considered confidential (e.g., data that is protected by legal privilege or by doctor–patient confidentiality, data that includes the content of individuals' non-public communications)?** All the data in STimage-1K4M is from public available source, we are not aware of such confidential information.
- **Does the dataset contain data that, if viewed directly, might be offensive, insulting, threatening, or might otherwise cause anxiety?** No.
- **Does the dataset identify any subpopulations (e.g., by age, gender)?** Not explicitly.
- **Is it possible to identify individuals (i.e., one or more natural persons), either directly or indirectly (i.e., in combination with other data) from the dataset?** No.
- **Does the dataset contain data that might be considered sensitive in any way (e.g., data that reveals race or ethnic origins, sexual orientations, religious beliefs, political opinions or union memberships, or locations; financial or health data; biometric or genetic data; forms of government identification, such as social security numbers; criminal history)?** No.

3. Collection Process

- **How was the data associated with each instance acquired?** We queried the GEO website using keywords "spatial transcriptomics", specifically targeting supplementary files including files in JPG, PNG, or TIFF formats. This search resulted in 856 datasets from 121 unique GEO studies. Additionally, we gathered 58 Visium and 4 VisiumHD datasets from 10X Genomics, complementing these with 233 slides manually collected from 10 additional studies.
- **What mechanisms or procedures were used to collect the data (e.g., hardware apparatuses or sensors, manual human curation, software programs, software APIs)?** We used rvest R package to gather download links for datasets in GEO. For other datasets, data were collected by human manual curation.
- **If the dataset is a sample from a larger set, what was the sampling strategy (e.g., deterministic, probabilistic with specific sampling probabilities)?** Not applicable, this dataset is not a sample from a larger set.
- **Who was involved in the data collection process (e.g., students, crowdworkers, contractors) and how were they compensated (e.g., how much were crowdworkers paid)?** Jiawen Chen, Wenrong Wu and Jinwei Zhang are involved in the data collection process. All of them are graduate students.

- **Over what timeframe was the data collected?** STimage-1K4M includes data generated from 2016-2024.
- **Were any ethical review processes conducted (e.g., by an institutional review board)?** No official ethical review processes were conducted.
- **Did you collect the data from the individuals in question directly, or obtain it via third parties or other sources (e.g., websites)?** The data were collected from websites.
- **Were the individuals in question notified about the data collection?** Not applicable, we collected the data from publicly available sources with no contact with individuals involved in the study.
- **Did the individuals in question consent to the collection and use of their data?** Not applicable, we collected the data from publicly available sources without contact with individuals involved in the study. We cite all the studies included in the dataset.
- **If consent was obtained, were the consenting individuals provided with a mechanism to revoke their consent in the future or for certain uses?** Not applicable, we collected the data from publicly available sources.
- **Has an analysis of the potential impact of the dataset and its use on data subjects (e.g., a data protection impact analysis) been conducted?** Not applicable, we collected the data from publicly available sources.

4. Preprocessing/cleaning/labeling

- **Was any preprocessing/cleaning/labeling of the data done (e.g., discretization or bucketing, tokenization, part-of-speech tagging, SIFT feature extraction, removal of instances, processing of missing values)?** Yes, the meta data like tissue type were cleaned manually. All code related to label cleaning is available in the GitHub repository `https://github.com/JiawenChenn/STimage-1K4M`.
- **Was the "raw" data saved in addition to the preprocessed/cleaned/labeled data (e.g., to support unanticipated future uses)?** Yes, all raw labels were saved.
- **Is the software that was used to preprocess/clean/label the data available?** Yes, all code related to label cleaning is available in the same GitHub repository.

5. Uses

- **Has the dataset been used for any tasks already?** No.
- **Is there a repository that links to any or all papers or systems that use the dataset?** Yes, all sources are available at `https://github.com/JiawenChenn/STimage-1K4M`.
- **What (other) tasks could the dataset be used for?** The dataset could be used for training self-supervised models to better understand histopathology and gene expression.
- **Is there anything about the composition of the dataset or the way it was collected and preprocessed/cleaned/labeled that might impact future uses?** Yes, we would recommend our format as the future data format release for ST data.
- **Are there tasks for which the dataset should not be used?** We are not aware of such task.

6. Distribution

- **Will the dataset be distributed to third parties outside of the entity (e.g., company, institution, organization) on behalf of which the dataset was created?** Yes, all the data are distributed under a permissible license for research-based use.
- **How will the dataset will be distributed (e.g., tarball on website, API, GitHub)?** The dataset is distributed on Huggingface: `https://huggingface.co/datasets/jiawennnn/STimage-1K4M`.
- **When will the dataset be distributed?** The dataset is released with a permissible license for research-based use.

- **Will the dataset be distributed under a copyright or other intellectual property (IP) license, and/or under applicable terms of use(ToU)?** The use of data will be under a permissible license for research-based use.
- **Have any third parties imposed IP-based or other restrictions on the data associated with the instances?** We are not aware of such restrictions.
- **Do any export controls or other regulatory restrictions apply to the dataset or to individual instances?** We are not aware of such restrictions.

7. Maintenance

- **Who will be supporting/hosting/maintaining the dataset?** The first author of the paper.
- **How can the owner/curator/manager of the dataset be contacted (e.g., email address)?** The first author and corresponding authors could be contacted using email listed in the paper or through GitHub.
- **Is there an erratum?** No.
- **Will the dataset be updated (e.g., to correct labeling errors, add new instances, delete instances)?** Yes, the dataset will be updated periodically to ensure data quality. We are also committed to continually expanding the dataset by adding new samples as they become available.
- **If the dataset relates to people, are there applicable limits on the retention of the data associated with the instances (e.g., were the individuals in question told that their data would be retained for a fixed period of time and then deleted)?** We are not aware of such limits.
- **Will older versions of the dataset continue to be supported/hosted/maintained?** All versions of the dataset will be available.
- **If others want to extend/augment/build on/contribute to the dataset, is there a mechanism for them to do so?** Not at this time.

## S2 Additional dataset information

STimage-1K4M is released at `https://github.com/JiawenChenn/STimage-1K4M` with metadata record also contained in this repository. The license for the data use is a permissible license for research-based use, which is described detailly on Huggingface: `https://huggingface.co/datasets/jiawennnn/STimage-1K4M`. All code related to this project is under MIT license.

## S3 Author Statement

The authors of the STimage-1K4M dataset bear full responsibility for the content and compliance of this project. All authors of this paper have confirmed the data license. The data is now hosted on ftp server and will be maintained by the first author of this paper.

## References

Gebru, T., J. Morgenstern, B. Vecchione, J. W. Vaughan, H. Wallach, H. D. Iii, and K. Crawford (2021). Datasheets for datasets. *Communications of the ACM 64*(12), 86–92.