# OpenReview forum: "STimage-1K4M: A histopathology image-gene expression dataset for spatial transcriptomics"
_NeurIPS.cc/2024/Datasets_and_Benchmarks_Track — NeurIPS 2024 Track Datasets and Benchmarks Poster_

### Official Review · Reviewer_8uje · 2024-07-19
**Spatial transcriptomics dataset**

**Rating:** 7
**Confidence:** 4

**Review:**

This contribution is significant to the field, as currently the training of multi-modal models with is a popular ongoing research topic. The dataset part of the paper is well-written, clear and to my impression is correct. On the other hand, I find the experiment description insufficient. Regarding the dataset, I would also expect more discussion and effort regarding batch effects.
I also find the access to the dataset a bit prohibitive.

Currently my mark is 6, but this paper is a valuable resource, I hope that authors can add updates and explanations on points raised above and I would be very happy to update my evaluation.

**Strengths:**

- **Diversity of the data**: authors collected data from many open sources and several ST techniques and many tissue types.
- **Data preprocessing**: Authors made a significant effort to align the data properly to ensure that the data presented is precise. I also appreciate the technical description in section 3 regarding the challenges and a high-level explanation on how authors were solving alignment issues.

**Additional Feedback:**

-

**Clarity:**

Paper is mostly well-written and very clear, except the section 5 on training experiment which is rather short and lacks details on data partition.

**Correctness:**

Dataset preparation steps seem to be correct.
I find it hard at the moment to evaluate correctness of experiment.

**Documentation:**

Authors provide a GitHub link, where the link for the dataset request can be found.
I find the access to the data quite restrictive. Why would a user need to sign contract? Paper does not discuss the policy and conditions on dataset access and usage.
Besides, I would advise to host the dataset somewhere else, for instance HuggingFace to ensure future access to it.

**Ethics:**

The dataset includes the data from the previously published research that involved human subjects. Authors also mention the limitation on over-representation of two tissue types in published data.
All in all, I don't see ethical concerns that would warrant further review.

**Limitations:**

Authors mention the bias towards two tissue types in current publications and I see that authors tried to collect a diverse dataset.

**Opportunities For Improvement:**

- **Description of the experiment**: I find section 5 hard to understand what is going on. Authors have fine-tuned the model, were the *samples from Maynard et al. (2021)* used for fine-tuning or only for evaluation afterwards?
- **Discussion on batch effects**: ST data suffers from batch effects between different experiment runs, labs and technologies. That is very important for the analysis and possibly for the dataset preprocessing. Unfortunately, that is barely mentioned in the paper and not at all discussed. Also, I am wondering if that would be possible to share batch-corrected \ normalized version of the dataset?
- **Access to the dataset**: I find the current way of accessing the data prohibitive and not sustainable. I think it could be hosted on a public platform, like HuggingFace.

**Relation To Prior Work:**

The related work is described and referenced well enough.

**Summary And Contributions:**

Authors contribute a spatial transcriptomics dataset: paired images and gene expression measurements. The dataset is diverse across ST technologies and tissue types. The dataset was preprocessed (aligning images and measurements).

---

> ### Author Rebuttal · Authors · 2024-08-15
>
> We sincerely thank the reviewer for their thorough, constructive and supportive feedback. Below, we provide point-by-point responses to address the comments and suggestions raised.
>
>
> **Opportunities For Improvement and Documentation**
>
> *Description of the experiment.*
>
> We appreciate the reviewer's feedback and understand the need for clearer exposition in Section 5. To enhance comprehension, we have added detailed explanations about our model. Specifically, we replaced the text encoder in both PLIP and CLIP with a single fully connected layer (gene expression encoder) and added a fully connected layer after the image encoder to align its dimensionality with that of the gene expression encoder, which is 32 in our case.
>
> To further improve clarity, we have incorporated a new figure depicting the structure of our proposed model (Figure R3 in "NewExperiment.pdf" avariable at https://github.com/JiawenChenn/STimage-1K4M/blob/main/aux/NewExperiment.pdf). For the fine-tuning/training of the CLIP and PLIP models, we have adhered to the same hyperparameter settings used during the original training of CLIP. To ensure clarity and ease of reference, we will present these hyperparameters in a detailed table within the appendix of our manuscript.
>
> In our training phase, we used all samples from Maynard et al. to do the fine-tuning/training. For the classification evaluation (Lines 264-273), we implemented a training, validation, and test split of 8:1:1 when training the linear classifier. And the evaluation on image representation learning is also conducted on all samples from Maynard et al. We have carefully revised all related text to avoid confusion. We hope these newly added context could help clarify the training process.
>
> *Discussion on batch effects.*
>
> We are grateful for the reviewer's insights and fully acknowledge the significance of batch effects in the integration of gene expression data. We previously addressed this issue in the discussion section, but to better highlight its importance, we have now included a t-SNE embedding of all samples from the Maynard et al. study, which comprises data from 12 human brain slides and demonstrates notable batch effects (Figure R4 in "NewExperiment.pdf"). Further commentary on this matter has also been added to the discussion section of our manuscript.
>
> Regarding the batch-corrected version of gene expression data, we acknowledge that finding a method that effectively addresses batch effects across different tissues, conditions, and species is challenging. While we are not currently aware of such a method, we are happy to share a normalized version of the gene expression data for convenient use by the research community. We would also be grateful for any recommendations of specific software or packages that the reviewer might suggest to help address these challenges.
>
> *Access to the dataset.*
>
> We thank the reviewer for the suggestion. We initially modeled our data release approach after Quilt-1M (NeurIPS 2023). Recognizing the potential access challenges associated with that method, we have now made our dataset available on HuggingFace under https://huggingface.co/datasets/jiawennnn/STimage-1K4M to ensure better accessibility.
>
> **Clarity:**
>
> We appreciate the reviewer's feedback, which prompted us to refine our manuscript further. We have added more detailed descriptions to better explain the experiments, including the new figures and tables mentioned previously. Regarding data partitioning, we did not perform any splitting for fine-tuning the CLIP and PLIP models. However, for the classification evaluation (Lines 264-273), we implemented a training, validation, and test split of 8:1:1 when training the linear classifier. We have made revisions throughout the relevant sections of the manuscript to eliminate any confusion surrounding these details.
>
> Thank you once again for your valuable feedback. We appreciate the opportunity to improve our work and remain open to any further discussion or suggestions you may have.

---

> > ### Comment · Reviewer_8uje · 2024-08-28
> >
> > Thank you for sharing the dataset and also adding other edits to the text, including the discussion of batch correction. I increase my score to 7.

---

> > > ### Author Rebuttal · Authors · 2024-09-01
> > >
> > > Thank you very much for your thoughtful feedback and for recognizing the improvements we made during the rebuttal. We greatly appreciate your decision to raise the score to 7 and your continued support.

---

### Official Review · Reviewer_8sup · 2024-07-24
**Review for STimage-1K4M**

**Rating:** 5
**Confidence:** 4

**Review:**

This quality of the dataset and the associated work is high, reflecting rigorous data collection and validation process. The dataset creation involves comprehensive coverage and validation processes. The paper is well-write and structured. Facilitating a clear understanding of the dataset's curations. The dataset is original since it integrates of gene expression data with histopathology images. It enables detailed studies of tissue architecture and cellular interactions, which are crucial for understanding disease mechanisms and developing targeted treatments.

**Pros**:
- High granular data
- Wide applicability
- Scalable dataset

**Cons**:
- Inconsistant formatting
- Clinical translation challenges
- Imbalanced organ composition

**Strengths:**

- Granular Data: The dataset’s granular approach with detailed gene expression data for sub-tile images allows for in-depth multi-modal analysis and could significantly advance computational pathology.
- Public Availability: The dataset and related code are publicly available, promoting transparency and enabling widespread use and verification by the research community.
- Potential Applicability: By providing data from a wide range of tissue types and species, though biased to brain and breast, it still offers valuable insights for researches focused on these and potentially other tissues included in the dataset.

**Additional Feedback:**

It would be better if author can explain the experimentations and evaluations.

**Clarity:**

Majority is well-written. However it's still strongly suggest to condense the introduction and leave more space for the discussion and experimentations.

**Correctness:**

Confusion in the experiments: it's bit confusion since fine-tune usually refer to supervised fine-tune. But the contrastive learning here doesn't necessarily refer to a supervised learning. Also for the linear probing, it's not clear what the model is predicting on (classification problem). It's would be better if author can clarify the experiment details.

**Documentation:**

Provided documentation are listed in the GitHub. However, there is no sustainable plan of hosting data. It would be better if author can explain it. It's also encouraging to provide the code which can directly generate the figure in the paper.

**Ethics:**

No ethical issue.

**Limitations:**

- Imbalanced tissue type: Despite the dataset including a variety of tissue types, the predominant focus on brain and breast tissues may not fully represent the complexities associated with other organs and diseases. This can limit the dataset’s applicability in studies targeting other specific tissues or diseases that are underrepresented.
- Link to the clinical usage: While the dataset provides a set of downstream tasks, translating these findings into clinical practice involves additional challenges, including extensive validation under clinical conditions, impact to the actual clinical usage like diagnosis or prognosis.

**Opportunities For Improvement:**

- Enhanced Documentation: Additional documentation or tutorials could help researchers better utilize the dataset, especially those new to spatial transcriptomics.
- Accessibility: Also consider hosting dataset in the place like huggingface to make sure it's sustainable and easy access.
- Expansion of Data Types: Incorporating more diverse types of gene expression data, such as from additional imaging-based ST technologies, could further enhance the dataset’s utility.

**Relation To Prior Work:**

It's clearly discussed with the previous work.

**Summary And Contributions:**

The paper introduces STimage-1K4M, a dataset designed to bridge the gap between histopathology images and gene expression data with spatial transcriptomics. It includes over 4 million pairs of sub-tile images and gene expressions derived from 1,149 images, aiming to enhance multimodal data analysis and CPATH applications. The dataset provides unprecedented granularity by offering detailed gene expression data for each sub-tile, which facilitates advanced research in CPATH and beyond.

---

> ### Author Rebuttal · Authors · 2024-08-15
>
> We sincerely thank the reviewer for their thoughtful and constructive feedback. Below, we provide point-by-point responses to address the concerns and suggestions raised.
>
> **Cons and limitations:**
>
> *Imbalanced tissue type.*
>
> We understand the reviewer's concern regarding the representation of tissue types in our dataset. However, the current distribution, with approximately 40\% brain and breast tissues, reflects the availability of publicly accessible data rather than a deliberate focus on certain tissues. We have made extensive efforts to gather as diverse a range of tissue types as possible, given the existing constraints in the field. We aim to regularly update the dataset as spatial transcriptomics technology becomes more widespread and commercially viable. This should facilitate the inclusion of a more diverse range of tissue types as more publicly available data emerges.
>
> *Link to the clinical usage.*
>
> Our dataset includes histology images that can be used to distinguish cancerous from non-cancerous regions and identify gene expressions pertinent to these areas. Furthermore, integrating our STimage-1K4M dataset with available medical images linked to clinical outcomes, such as survival rates, offers the potential to deepen our understanding of these correlations. However, we acknowledge that bridging the gap to clinical utility remains a significant challenge in the omics fields. We will incorporate these points into our discussion section.
>
> *Clarification on fine-tune.*
>
> We thank the reviewer for pointing out that the potential confusing term "fine-tune" in our manuscript, as it is commonly associated with supervised learning context. We will clarify this usage at its first occurrence by stating: "In this manuscript, we use the term 'fine-tune' to describe adjustments within a contrastive learning framework." We are open to modifying this term if the reviewer suggests an alternative.
>
> *Clarification on experiments.*
>
> We appreciate the reviewer’s feedback on linear probing experiments. In the current version of the paper, we conduct two experiments. The first is a supervised classification task, where we used a penalized logistic regression model with image embeddings to predict brain layers (WM, L1-L6, Figure 3c). The second involves image representation learning, where brain layer labels directly assess the image embeddings without further classifier training. We have revised the manuscript to clearly explain these points in response to the reviewer's suggestions.
>
> *Inconsistant formatting.*
>
> We apologize that we had a hard time understand this comment. We hope that the reviewer could clarify this and we are happy to address the concerns.
>
>
> **Opportunities for improvement and Documentation:**
>
> *Enhanced Documentation.* We are grateful for the reviewer's suggestion. In response, we have enriched the resources available to researchers new to spatial transcriptomics (ST). Our website now hosts two new documents: one introduces ST technologies, and the other serves as a quick start guide for ST analysis, accessible via https://jiawenchenn.github.io/STimage-1K4M/docs/09_for_new_to_ST. We have included code detailing how to convert STimage-1K4M data into the widely-used AnnData format. This enhancement will facilitate the use of our dataset with popular ST analysis packages, aiming to simplify the entry process for newcomers to the field.
>
> We have also updated our documentation to incorporate descriptions of the two new experiments and have included all the necessary code to generate the figures presented in our work, for reproducibility. We are committed to further improving our documentation and are open to including any additional resources or materials that the reviewer might suggest.
>
> *Accessibility.*
>
> We thank the reviewer for the suggestion. We initially modeled our data release approach after Quilt-1M (NeurIPS 2023). Recognizing the potential access challenges associated with that method, we have now made our dataset available on HuggingFace under https://huggingface.co/datasets/jiawennnn/STimage-1K4M to ensure better accessibility.
>
>
> *Expansion of Data Types.*
>
> Currently, STimage-1K4M focuses on sequencing-based technologies due to the availability of associated image data, while its generally challenging to find image sources for imaging-based technologies. We recognize that integrating imaging-based technologies could enhance the dataset’s utility. To address this, we will mention a recently curated dataset, SpatialCorpus-110M, which focuses on imaging-based ST data including technologies like Merfish, Xenium, ISS, and CosMx, covering approximately 54 million cells. This dataset could complement ours well, and we plan to discuss it in the paper to inform readers. Moreover, we are committed to ongoing updates that will include new types of ST data that combine both image and gene expression information.
>
> *Code for generating figures.*
>
> We thank the reviewer for the suggestion. We have now included all code to generate figures at https://jiawenchenn.github.io/STimage-1K4M/docs/13-code-for_figs.

---

> > ### Author Rebuttal · Authors · 2024-08-15
> >
> > **Clarity:**
> >
> > We thank the reviewer for the comments. We have revised the experiment part with more detailed information as in the response above. In addition, we have expanded our experiments to address all tasks mentioned in our Section 4: Popular tasks using ST images. In response, we have expanded our experimental scope as detailed in "NewExperiment.pdf" avariable at https://github.com/JiawenChenn/STimage-1K4M/blob/main/aux/NewExperiment.pdf. The additional experiments now include a prediction task (Figure R1) and a nuclei segmentation task (Figure R2), improving the comprehensiveness of our study.
> >
> >
> > For the gene expression prediction ask, we aim to evaluate the ability of zero-shot image embeddings from CLIP, PLIP, and UNI in predicting gene expression. Using the same HER2+ breast cancer dataset, we trained a ridge regression model on slide H1 and tested it on slide H2, sourced from the same patient. Specifically, we assessed the ability to predict the expression of ERBB2, a marker gene for HER2+ breast cancer typically overexpressed in cancerous regions. Surprisingly, PLIP exhibited the best performance in predicting ERBB2 expression.
> >
> >
> > For the nuclei segmentation/deconvolution task, a traditional approach in spatial transcriptomics (ST) involves using spot-level data—comprising gene expression, spatial coordinates, and imagery, often alongside an external scRNA-seq reference—to infer cell type proportions for each spot (see PMID35753702 for a summary of ST deconvolution methods). The inferred cell type proportion could be considered an alternative benchmark for complementing the lack of nuclei and cell type annotation which required extensive manual effort. With STimage-1K4M, we can benchmark the performance of cell segmentation methods using the inferred cell type proportion.
> >
> > Specifically, we assessed the performance of the CellViT-256 and CellViT-SAM models in segmenting Neoplastic cells and compared these to the Cancer epithelial cell proportions inferred by RCTD on HER2+ breast cancer data. Visual inspections confirm comparable performance in cell segmentation. Notably, the CellViT-SAM model tends to identify more cells as Neoplastic, whereas the CellViT-256 model leans towards classifying cells as connective. We evaluated the agreement between segmented and inferred cell types using Pearson’s correlation, finding that CellViT-256 shows a higher agreement level (0.37 vs. 0.34 for CellViT-SAM).
> >
> >
> > We will incorporate all newly conducted experiments into the manuscript to provide a comprehensive overview of how to use STimage-1K4M for benchmarking. We have also enriched the discussion section of our manuscript by including more detailed discussion about the potential for clinical translation and providing insights into other imaging-based datasets.
> >
> > Thank you once again for your valuable feedback. We remain open to further discussion and look forward to continuing to improve our work.

---

> > > ### Comment · Reviewer_8sup · 2024-08-31
> > >
> > > Thank you very much for your response! I think I am satisfied with the revision and all comments you provided. I intend to increase my score to 6.

---

> > > > ### Author Rebuttal · Authors · 2024-09-01
> > > >
> > > > Thank you very much for your thoughtful feedback during the rebuttal. We are glad that you are satisfied with the revisions and comments we provided. We appreciate your intention to raise the score to 6 and your continued support.

---

### Official Review · Reviewer_LBXw · 2024-07-24
**Good dataset**

**Rating:** 6
**Confidence:** 2

**Review:**

Pros:

1. Very useful dataset with paired histopathology images and gene expression.

2. The paper mentioned several popular downstream tasks of the dataset that are important and impactful.

3. Experiments showed that the collected data can improve CLIP and PLIP performance on computational pathology data through fine-tuning on this dataset.

Cons:

1. The experiments are not comprehensive enough. They did not cover all modalities in the dataset, and did not cover all mentioned use tasks of the dataset.

**Strengths:**

Very useful dataset for computational pathology. The author listed several important tasks that the dataset can be used to address, including gene expression prediction and super-resolution, patch representation learning, and cell segmentation. The authors also demonstrated that the dataset can improve CLIP and PLIP performance on computational pathology tasks through fine-tuning.

**Additional Feedback:**

N/A

**Clarity:**

The paper is mostly clear, although its technical details about ST technologies in the data collection section can be challenging for readers who are not already familiar with ST.

**Correctness:**

The dataset seems to be correctly constructed. The experiment seems to be a bit lacking, not covering all modalities in the dataset and all claimed downstream tasks of the dataset (see "opportunities for improvement" section for details).

**Documentation:**

Documentation is adequate.

**Ethics:**

No ethical concern.

**Limitations:**

For the experiments suggested in "opportunities for improvement" above, if some of the evaluations are impossible to perform, perhaps the authors can mention them in the limitations section explaining why they cannot be conducted.

**Opportunities For Improvement:**

The experiments can be more comprehensive. Currently the experiments seem to only evaluate image representation learning as well as linear probing classification results; however, the author mentioned many possible tasks that STimage-1M4K can be used for, such as cell segmentation and gene expression prediction or super-resolution, and there is no experiments on these tasks. Furthermore, the experiments seem to only involve the image and gene expression modalities, so it is unclear whether the other data in Figure 1(a) (such as spot location, sub-pathology, text like abstract/keywords/summary) are useful.

**Relation To Prior Work:**

Discussion is adequate.

**Summary And Contributions:**

In this paper, the authors curated a new dataset called STimage-1K4M, which contains paired histopathology image - gene expression data. The dataset is the biggest in its type, consisting of over 4 million pairs. The collected data is collected with three ST technologies, and the data is pulled from public available sources like GEO as well as from additional sources from a large number of existing studies. The authors evaluated CLIP, PLIP, and UNI performance on the new dataset, and demonstrated that fine-tuning CLIP and PLIP on the dataset improves their performance.

---

> ### Author Rebuttal · Authors · 2024-08-15
>
> We sincerely thank the reviewer for their valuable and insightful feedback. We have carefully considered your comments and have made revisions accordingly. Below, we provide detailed responses to each of the points raised.
>
> **Cons, Opportunities For Improvement, Correctness and Limitations:**
>
> Thanks for the suggestions. We have expanded our experiments to address all tasks mentioned in our Section 4: Popular tasks using ST images. In response, we have expanded our experimental scope as detailed in "NewExperiment.pdf" avariable at https://github.com/JiawenChenn/STimage-1K4M/blob/main/aux/NewExperiment.pdf. The additional experiments include a gene expression prediction task (Figure R1) and a nuclei segmentation task (Figure R2) and, improving the comprehensiveness of our study.
>
> For the gene expression prediction task, we aim to evaluate the ability of zero-shot image embeddings from CLIP, PLIP, and UNI in predicting gene expression. Using the same HER2+ breast cancer dataset, we trained a ridge regression model on slide H1 and tested it on slide H2, sourced from the same patient. Specifically, we assessed the ability to predict the expression of ERBB2, a marker gene for HER2+ breast cancer typically overexpressed in cancerous regions. In this task, PLIP exhibited the best performance in predicting ERBB2 expression.
>
> For the nuclei segmentation/deconvolution task, a traditional approach in spatial transcriptomics (ST) involves using spot-level data—comprising gene expression, spatial coordinates, and imagery, often alongside an external scRNA-seq reference—to infer cell type proportions for each spot (see PMID35753702 for a summary of ST deconvolution methods). The inferred cell type proportion could be considered an alternative benchmark for complementing the lack of nuclei and cell type annotation which required extensive manual effort. With STimage-1K4M, we can benchmark the performance of cell segmentation methods using the inferred cell type proportion.
>
> Specifically, we assessed the performance of the CellViT-256 and CellViT-SAM models in segmenting Neoplastic cells and compared these to the Cancer epithelial cell proportions inferred by RCTD on HER2+ breast cancer data. Visual inspections confirm comparable performance in cell segmentation. Notably, the CellViT-SAM model tends to identify more cells as Neoplastic, whereas the CellViT-256 model leans towards classifying cells as connective. We evaluated the agreement between segmented and inferred cell types using Pearson’s correlation, finding that CellViT-256 shows a higher agreement level (0.37 vs. 0.34 for CellViT-SAM).
>
>
>
> We appreciate the reviewer's suggestion to utilize all modalities in our dataset. We have made significant effort to collect additional data such as abstracts, titles, and other textual information, making ours the first dataset to include these elements for utilization. However, the application of this textual data remains an open challenge. In this study, we utilized spot-level image and gene expression data. However, the textual information is at the slide level, and we currently lack a method to effectively integrate this data into our analyses. We will acknowledge this in our limitations section and welcome further experimental suggestions from reviewers.
>
>
> **Clarity:**
>
> We thank the reviewer for the comment. For a better understanding of the ST field, we will add a discussion about several recently published review papers including "Unlocking the power of spatial omics with AI" (PMID 39122938) and "Transformers in single-cell omics: a review and new perspectives" (PMID 39122952), in the introduction.
>
> We thank the reviewer once again for their constructive feedback, which has significantly enriched our submission. We believe presenting our dataset at a high-impact conference like NeurIPS will attract attention from experts in computer vision and multi-modality modeling, potentially leading to innovative solutions for the ST field. We are committed to keep improving our work and remain open to further feedback and discussion.

---

> > ### Comment · Reviewer_LBXw · 2024-08-28
> >
> > Thank you very much for your response! My review remains positive.

---

> > > ### Author Rebuttal · Authors · 2024-09-01
> > >
> > > Thank you very much for your helpful feedback during the rebuttal and for maintaining your positive review. We truly appreciate your continued support.

---

### Official Review · Reviewer_cGjP · 2024-07-25
**STimage-1K4M**

**Rating:** 7
**Confidence:** 4
**Clarity:** It is easy to read and well written.

**Review:**

See strengths and weaknesses.

**Strengths:**

- **Novel dataset:** STimage-1K4M addresses a significant gap in existing medical image-text datasets providing paired histopathology images and gene expression data at a granular level. The authors detail their meticulous approach for collecting and processing the data, ensuring consistency and quality across diverse sources.

- **Scale and diversity:** The dataset covers a wide range of species and tissue types, making it valuable for various research applications in computational pathology and spatial transcriptomics. High-dimensional gene expression data paired with images enables more sophisticated multi-modal analyses and machine learning applications.

- **Inclusion of pathologist annotations:** The addition of expert annotations for a subset of slides provides valuable ground truth data for model evaluation.

- **Potential tasks:** I appreciate the details regarding the potential tasks that can be addressed with the dataset, including evaluation of fine-tuned CLIP and PLIP to validate the improved representation learning ability.

**Additional Feedback:**

- Appendix (L.1045 - 1047) should be left aligned.

- Appendix (L.1033 - 1034): This needs further clarification, downsampling to $32 \in D$ latent dimension could be the reason for marginal gains between the zero-shot and fine-tuned models. Was the text-encoder replaced with only one fully-connected layers? There seems to be an inconsistency with the details mentioned in the main paper and that of the appendix.

- While the  collection of the dataset is a significant effort. The evaluations do not clearly underscore the gains across each potential task and it not clear which downstream tasks image classification were evaluated via linear probing.

**Correctness:**

The authors detail the collection processes clearly and include several baselines to validate the potential tasks.

**Documentation:**

Details included in supplementary material and external link.

**Ethics:**

None.

**Limitations:**

- **Limited fine-tuning experiments**: The study only evaluates fine-tuning performance using the human brain dataset as a benchmark, which may not fully represent the dataset's potential across different species and tissue types. In addition, details regarding actual fine-tuning and hyper-parameter selection for finetuned CLIP/PLIP are limited.

- **Batch effects**: The authors acknowledge significant batch effects across datasets, which could limit the generalizability of models trained on this data.  However, they acknowledge this and the need for species-specific analysis due to differences in gene names across species.

- **Potential bias and utility**: The dataset may have inherent biases due to the focus of existing ST publications on certain tissue types (e.g., brain and breast cancer), which could influence downstream analyses.

**Opportunities For Improvement:**

- Expanding the fine-tuning experiments to include multiple species and tissue types may better demonstrate the dataset's broad applicability.

- Moreover, while the performance of UNI is quite higher than the CLIP/PLIP models, the gains of the zero-shot and image-gene learned representation evaluation are marginal. Several clarifications on training and the why a single layer was used are required.
- Explore more advanced multi-modal architectures that can better leverage the high-dimensional gene expression data alongside image information.

**Relation To Prior Work:**

.Yes.

**Summary And Contributions:**

This study introduces STimage-1K4M, a comprehensive dataset that pairs histopathology images with gene
expression data from spatial transcriptomics (ST) technologies. As opposed to existing medical image-text
datasets that lack sub-granular annotations for sub-regions, leveraging ST data provides high dimensional
gene expression for tissue sub-regions enabling more detailed multi-modal analysis. The dataset includes
1,149 ST slides from three leading technologies, resulting in 4,293,195 image-gene expression pairs
across 10 species and 50 tissue types. The authors demonstrate the dataset's potential for representation
learning by fine-tuning pre-trained vision-language models (CLIP and PLIP) image encoders and replace the
text-encoder with a gene-encoder. Linear probing evaluation shows improved performance in image
classification and clustering tasks. Details on data collection and potential biases in the dataset
are highlighted.

---

> ### Author Rebuttal · Authors · 2024-08-15
>
> We greatly appreciate the time and effort you've taken to review our work. Below, we provide point-by-point responses to address your helpful comments and suggestions:
>
> **Opportunities For Improvement:**
>
> *Expanding fine-tuning to multi-species/tissues.*
>
> We agree that expanding our fine-tuning experiments to multiple species and tissue types will improve the relevance and applicability of our dataset. Based on your suggestions, we have expanded our analysis in two ways: fine-tuning our model across more tissue types (human brain, human breast, mouse brain), and extending testing to include all available annotated datasets (currently limited to human brain, human breast, mouse brain, mouse lung, human prostate and human kidney). We have added the expanded experiments to our Appendix. See Figure R5 in the newly added "NewExperiment.pdf" document available at https://github.com/JiawenChenn/STimage-1K4M/blob/main/aux/NewExperiment.pdf for more details.
>
> In our study, we initially focused on fine-tuning with samples exclusively from the human brain dataset provided by Maynard et al. (2021) (Figure 3). Using data from similar biological conditions helps control for batch effects and other potential biases that may arise from integrating diverse data sources. However, it is noteworthy that models fine-tuned with diverse tissue types generally sometimes performed worse than their non-fine-tuned counterparts. This highlights the significant challenge of integrating gene expression and imaging data across multiple tissues and species, which is an issue that remains unresolved within the field.
>
> We acknowledge the limitations of our current methodologies and do not yet have a perfect solution to these integration challenges. Nonetheless, we believe that the release of our STimage-1K4M dataset will inspire researchers worldwide to develop innovative and effective strategies to leverage complex gene expression data. We are eager to see the community's creative approaches to this pressing problem.
>
> To further demonstrate the versatility of our dataset, we have expanded our experimental scope as outlined in Section 4 of our manuscript, titled "Popular Tasks Using ST Images." Details of these additional experiments are provided in the "NewExperiment.pdf". The expanded experiments include tasks such as nuclei segmentation (Figure R2 in "NewExperiment.pdf") and gene expression prediction (Figure R1 in "NewExperiment.pdf"), enhancing the overall comprehensiveness of our study. We hope these efforts will illustrate the broader potential applications of our dataset to the reviewer.
>
>
> *Clarification on the different performance on UNI in classification and representation learning.*
>
> We appreciate the reviewer's observations on the performance differences between UNI, CLIP, and PLIP models on different tasks. To clarify, we evaluated these models in two contexts: supervised classification, where we utilized a penalized logistic regression with image embeddings as inputs to predict brain layers (WM, L1-L6); and image representation, where the same brain layer labels were used to directly score the image embeddings without additional classifier training. The differences in performance between these models may be attributed to several factors. For instance, the inherent design and optimization of each model might favor certain types of data representations or tasks. Furthermore, the fine-tuned CLIP and PLIP models exhibit sparser image representations (as shown in Figure 3d), which could complicate their ability to predict brain layers in a supervised setting. However, this sparsity might enhance their performance in unsupervised clustering evaluations. We will add this discussion to the experiment part in the revised manuscript.
>
> *Clarification on single-layer.*
>
> We interpret the reviewer's comment as requesting further clarification on our choice to use a single-layer gene expression encoder. We opted not to explore various model structures extensively in this study because developing a novel, robust model architecture remains an open challenge within the field. We acknowledge that this is a limitation of our current approach and will include this point in the limitations section of our manuscript.
>
> *Explore more advanced multi-modal architectures.*
>
> We thank the reviewer for highlighting this important aspect. The development of optimal multi-modal architectures for integrating image and gene expression data remains an unresolved challenge in the spatial transcriptomics (ST) field. Commonly, a contrastive learning framework is employed, and in this vein, we have expanded the current employed smaller-scale models (such as the setting in PMID 38352580) to include large vision model image encoders. Despite these efforts, identifying superior multi-modal architectures remains an open question.
>
> We hope that the release of our dataset will inspire other researchers to explore innovative solutions in this area. We are also open to expanding the discussion in our paper to include potential advanced multi-modal architectures. Any insights or suggestions from the reviewer would be greatly appreciated and would help enrich the discussion, fostering further exploration and development within this promising field.

---

> > ### Author Rebuttal · Authors · 2024-08-15
> >
> > **Limitation and Additional Feedback:**
> >
> > *L.1045 - 1047 formatting.*
> >
> > We thank the reviewer for the careful check. We have corrected this part.
> >
> > *Fine-tune detail and hyperparameter choices.*
> >
> > We appreciate the reviewer's observations. Yes, you are right. We replaced the text encoder in PLIP and CLIP with a single fully connected layer. We also add a fully connected layer after the image encoder to make the dimension same as the gene expression encoder, which in our case is 32. We have thoroughly revised the relevant sections of our manuscript to enhance clarity. Additionally, we have incorporated a new figure that better illustrates the structure of our proposed model (Figure R3 in "NewExperiment.pdf").
> >
> > For the fine-tuning of the CLIP and PLIP models, we used the same hyperparameter settings as the original training of CLIP (batch size = 1024, epoch =15, lr=5e-5, betas=(0.9,0.98), eps=1e-6, weight\_decay=0.2). We will present these hyperparameters in a detailed table in the appendix for clarity.
> >
> > *Further clarification on model performance.*
> >
> > For evaluation, our methodology involved using a 32-dimensional latent variable for the fine-tuned models and a 512-dimensional representation for zero-shot image embeddings across both classification and image representation learning tasks. The choice of a 32-dimensional space for the fine-tuned models was driven by our contrastive models, which construct contrastive loss based on this 32-dimensional latent layer. We have revised all relevant sections in both the main manuscript and the supplementary material to clarify these details.
> >
> > Furthermore, we acknowledge the reviewer's concern that the sub-optimal performance of the fine-tuned models could stem from the reduction to a 32-dimensional representation. To address this, we conducted evaluations using both 32-dimensional and 512-dimensional spaces and found the performance to be comparable (Figure R5, R6 in "NewExperiment.pdf"). We have included these findings in the Appendix to provide a comprehensive overview of our evaluation results and to substantiate our methodology.
> >
> >
> > *Clarification on linear probing.*
> >
> > We are grateful for the reviewer's comment. The image classification task aimed to assess clustering performance in a supervised manner (Figure 2c). We also have expanded our experimental section to include gene expression prediction evaluation (Figure 2a) and deconvolution/nuclei segmentation evaluation (Figure 2d), ensuring a more comprehensive evaluation across different tasks.
> >
> > To improve clarity, we will reorder the potential tasks listed in Section 4 to align with the sequence of experiments presented. We hope these adjustments address your concerns effectively.
> >
> >
> > Thank you again for the helpful feedback. We remain open to further experiments, clarifications, and discussions.

---

> > ### Comment · Reviewer_cGjP · 2024-08-31
> > **Final Review**
> >
> > I would to thank the authors for carefully considering each concern. I carefully reviewed the rebuttal and changes made, which are both satisfactory and helpful in better contextualizing the core contributions of this work. Moreover, the educational resource and documentation is equally appreciated. In this regard, I am pleased to raise my score to 7 as I am of the view the community will benefit from this resource.

---

> > > ### Author Rebuttal · Authors · 2024-09-01
> > >
> > > Thank you very much for your thoughtful and supportive feedback during this rebuttal phase. We are glad that the changes and clarifications were satisfactory and helpful in contextualizing our work. We also appreciate your decision to raise the score from 5 to 7, recognizing the value of our contributions.

---

### Author Rebuttal · Authors · 2024-08-15

We are grateful for the extensive feedback provided by the reviewers and have made significant improvements to our manuscript in response. Below, we discuss some of the key points and recurring comments in this global response. Additionally, we address each reviewer's specific comments and questions point by point in the individual rebuttals.

**Expanded Fine-Tuning and Evaluation (Reviewer cGjP)**

We have broadened the scope of our fine-tuning experiments to include a wider range of tissue types and species. Correspondingly, our evaluation now also encompasses a more diverse set of tissues (for both training and testing datasets), to demonstrating the broad applicability of our dataset. The results can be found in Figure R5,R6 in the "NewExperiment.pdf" document available at https://github.com/JiawenChenn/STimage-1K4M/blob/main/aux/NewExperiment.pdf.

**Enriched Experiments (Reviewers cGjP, LBXw)**

We have added two new experimental categories: gene expression prediction and cell segmentation, to show the versatility of our dataset in addressing varied biological questions. The results can be found in Figure R1 and R2 in the "NewExperiment.pdf" document.

**Dataset Accessibility (Reviewers 8sup, 8uje)**

 To facilitate easier access to our dataset, we have hosted it on Hugging Face ( https://huggingface.co/datasets/jiawennnn/STimage-1K4M) for easy access.

**Expanded Discussion on Challenges and Limitations (Reviewers cGjP, LBXw, 8sup, 8uje)**

We have enriched the discussion section by elaborating on the challenges related to batch effects, the current limitations of our experiments, and the potential for clinical translation. This helps clarify our findings and highlights the ongoing challenges in the field.

**Enhanced Documentation (Reviewer 8sup)**

We have augmented our documentation to include resources for researchers who are new to spatial transcriptomics. This includes an introduction to the technology and methods in the field, as well as a quick-start guide on how to use the STimage-1K4M dataset for standard ST analysis (both available at https://jiawenchenn.github.io/STimage-1K4M/docs/09_for_new_to_ST). Additionally, we have made all code for reproducing the figures in our paper available to ensure transparency and reproducibility. All the documents can be accessed via https://jiawenchenn.github.io/STimage-1K4M/docs/01-make-meta.

**Detailed Documentation of Model Structure and Experiments (Reviewer cGjP, 8sup, 8uje)**

We have updated the experimental documentation to include more detailed information about the model we propose, complete with a new figure that clarifies its structure (Figure R3 in "NewExperiment.pdf" document). We also have enriched the documents related to training data partition, hyper-parameter choices for fine-tuning, the evaluation process, and other additional experimental details.

We have revised our manuscript thoroughly to address the points raised by the reviewers and to eliminate any potential confusion. We sincerely appreciate the considerable effort the reviewers have put into evaluating our work, which has significantly improved our work. We remain open to incorporating any additional experiments, documents, clarifications, or discussions.

---

### Decision · Program_Chairs · 2024-09-26

**Decision:**

Accept (Poster)

**Comment:**

**Submission #1305**: "STimage-1K4M: A histopathology image-gene expression dataset for spatial transcriptomics"

**Recommendation: Accept (Poster)**

**Comment/Note**: The overall contribution is not entirely dissimilar to the HEST-1k submission (#637). Both take a slightly different approach, and in my (admittedly non-expert) opinion both are above the publication threshold, with this submission being more suitable for a poster presentation, as far as I can tell.

**Score note**: one reviewer responded with their intention to increase their score from 5 (marginally below threshold) to 6 (marginally above threshold), which they likely forgot, which would have brought the total average (of primary reviewer scores) to 6.5, with all final scores being 6 or higher.

**Summary**: The authors present STimage-1K4M, with differently resolved spatial transcriptomics (ST) "annotations" across 4 million "tiles" (spatial sub-locations with central coordinates within the 1,149 images). Given the different noise/quality (tile size/amount of tissue that went into the transcriptomics process per tile), some care will have to be taken when applying this dataset to future endeavors. Overall, the work seems very solid, despite some remaining shortcomings (which the authors acknowledge in the rebuttals).

The reviewers in particular mention the novelty, scale, and diversity of the data as strengths, together with the very accessible format and presentation in the manuscript.